Corrected: Author correction

# Infralimbic cortex is required for learning alternatives to prelimbic promoted associations through reciprocal connectivity

Arghya Mukherjee [1] & Pico Caroni [1]

Prefrontal cortical areas mediate flexible adaptive control of behavior, but the specific contributions of individual areas and the circuit mechanisms through which they interact to modulate learning have remained poorly understood. Using viral tracing and pharmacogenetic techniques, we show that prelimbic (PreL) and infralimbic cortex (IL) exhibit reciprocal PreL↔IL layer 5/6 connectivity. In set-shifting tasks and in fear/extinction learning, activity in PreL is required during new learning to apply previously learned associations, whereas activity in IL is required to learn associations alternative to previous ones. IL→PreL connectivity is specifically required during IL-dependent learning, whereas reciprocal PreL↔IL connectivity is required during a time window of 12–14 h after association learning, to set up the role of IL in subsequent learning. Our results define specific and opposing roles of PreL and IL to together flexibly support new learning, and provide circuit evidence that IL-mediated learning of alternative associations depends on direct reciprocal PreL↔IL connectivity.

[1] Friedrich Miescher Institut, Basel CH-4058, Switzerland. Correspondence and requests for materials should be addressed to P.C. (email: caroni@fmi.ch)

Top-down control through prefrontal cortex (PFC) is believed to link internal goals to perception, thought and action in order to promote flexible behavior and guide learning[1]. Studies in humans, monkeys, and rodents have revealed how pyramidal neurons in different prefrontal areas are tuned to distinct aspects of goal-oriented behavior, including the detection of expected and unexpected information, of the value of rewards, information and actions, and of options selected and not selected[2–5]. From these studies, the view has emerged that prefrontal areas might, together, inform circuitry involved in learning outside these areas about the changing value of information and possible actions, e.g., by keeping this information in working memory to direct attention and goal-oriented behavior[1,6,7]. Testing these predictions and understanding the roles of different subdivisions of prefrontal cortex in learning and memory, as well as understanding the underlying circuit mechanisms, might be achieved through area- and circuit-specific loss and gain-of-function experiments during learning.

PreL and IL are two major medial PFC areas in rodents thought to mediate control of goal-oriented behavior[8–10]. Both PreL and IL neurons exhibit dense connectivity with brain areas such as striatum, nucleus accumbens, amygdala[11,12] and hippocampus, consistent with roles in top-down control of learning and memory[13–15]. Neuronal activity in human dorsolateral PFC (a prefrontal area related by some studies to rodent medial PFC) was correlated to the reward value of action choices[10,16,17]. Furthermore, PreL neurons were tuned to the value of goals, e.g., in goal-directed spatial navigation[8,9], whereas IL neurons appear to be tuned to alternative choices[18]. In rodents, lesion studies have suggested that medial PFC (including PreL and IL) is critically important for flexible behavior[18–22]. IL was shown to be critically important in extinction of Pavlovian fear conditioning and in controlling addiction[23–30]. Furthermore, IL has been implicated in the development of habits[31–33], and in the control of pain, impulsivity and depression[34–36]. Taken together, neuronal tuning and lesion studies have suggested that rodent PreL and IL might have complementary and perhaps opposite roles in goal-oriented flexible learning[24,30,37–39]. Comparable important roles for flexible behavior as those assigned to rodent PreL and IL, have been assigned to primate PFC. These findings have raised the question of how PreL and IL might interact to influence learning, and what might be the possible underlying circuit mechanisms. Elucidation of these questions might provide insights into how distinct but functionally related prefrontal cortical areas might together support flexible learning and memory.

To investigate whether and how PreL and IL might exert complementary and possibly opposite roles in learning, we carried out local PreL or IL silencing experiments during learning or its recall, and traced and functionally targeted connectivity between these two prefrontal areas using specific viral vector mediated tracing and pharmacogenetic silencing. Using intra/extradimensional set-shifting (IEST) tasks, as well as fear conditioning and extinction protocols, we provide evidence that activity in PreL is required during new learning to promote application of previously learned associations, whereas activity in IL is required to learn alternative associations. Neither activity in PreL nor in IL was required to recall learned associations when no new learning was involved. We further show that the role of IL to promote alternative learning specifically depends on IL→PreL connectivity during learning, whereas reciprocal PreL→IL and IL→PreL layer 5/6 connectivity is specifically required during a time window 12–14 h after learning of a particular association to set up the role of IL to learn alternative associations. Taken together, our results define distinct and opposite roles of PreL and IL specifically to support new learning, and provide circuit-level evidence that IL mediates learning of associations alternative to those supported by PreL through direct reciprocal PreL-IL connectivity.

## Results

**Reciprocal PreL/IL layer 5/6 connectivity**. As a first step toward investigating circuit mechanisms underlying possible interactions between PreL and IL in learning, we determined whether these adjacent prefrontal areas were directly connected with each other. We delivered a Cre-expressing retrograde CAV2[40] specifically to PreL (or IL) and AAV8 expressing Cre-dependent DREADDs[41] and mCherry for visualization and activity manipulation of dual-infected neurons in IL (or PreL) (Fig. 1a, Supplementary Fig. 1). Twelve to Fourteen days after these viral deliveries, we analyzed brain sections for marked projection neurons. We detected prominent reciprocal layer 5/6 connectivity between PreL and IL (Fig. 1a). Specifically, neurons projecting from IL to PreL were mainly located in layer 6 ($59.1 \pm 3.6\%$ of total NeuN + neurons, $n = 5$ animals), followed by layer 5 ($33.2 \pm 7.3\%$) ($11.1 \pm 1.7\%$ of total GABAergic[42]), and neurons projecting reciprocally from PreL to IL were located to comparable extents in layer 6 ($48.1 \pm 5.5\%$ of total) and layer 5 ($45.3 \pm 3.7\%$) ($4.7 \pm 3.2\%$ of total GABAergic) (Fig. 1b).

To determine the cortical layer(s) in which IL→PreL (or PreL→IL) terminate within the respective other cortical area, we applied combinatorial infection of CAV2-Cre and AAV9-flex-synaptophysin-Myc to infect these neurons, visualizing their synaptic terminals (Fig. 1c). We found that axons of IL→PreL or of PreL→IL projection neurons terminate in layer 6 and layer 5 of PreL, respectively (IL). In addition, collaterals of these same axons targeted basolateral amygdala (BLA) (Fig. 1c), but not a number of additional potentially relevant target regions, including nucleus accumbens (Fig. 1c; we detected numerous GFP + axons, but failed to detect synaptophysin-Myc + presynaptic terminals in this area), hippocampus or primary motor cortex M1 (Fig. 1c). For nucleus accumbens, it is possible that weak or spatially highly restricted presynaptic terminals by IL→PreL and/or PreL→IL projection neurons went undetected in our experiments. Alternatively, different neurons might account for functional projections between PreL/IL and nucleus accumbens.

**Shift-learning: PreL to apply rules and IL to learn alternatives**. We then investigated specific requirements for PreL and IL in learning and memory by transiently and bilaterally silencing specifically PreL or IL at acquisition or recall of different forms of learning. To achieve local silencing of PreL or IL, we pharmacogenetically activated local parvalbumin (PV) interneurons, thereby effectively silencing local principal neurons[43–46]. To this end, we injected an AAV expressing a Cre-dependent activating ligand-gated ion channel construct (PSAM-5HT3; this engineered hybrid channel construct does not respond to the endogenous neurotransmitters ACh or 5HT, and the artificial ligand PSEM308 (short PSEM) does not displace 5HT from endogenous 5HT3 receptor)[43] in a PV-Cre mouse line. The neurons expressing the channel could be detected by labeling with α-Bungarotoxin-Alexa488, and activated by i.p. delivery of the corresponding ligand PSEM. Specific targeting of IL was achieved under stereotactic guidance through slightly oblique orientation delivery angles. This was done in order to reach IL without contacting PreL, thereby minimizing drug and virus spread to PreL (Fig. 2a, Supplementary Fig. 1; Methods). Silencing specificity and effectiveness were verified through analysis of cFos induction in layer 2/3 neurons of PreL and IL 90 min after acquisition (Supplementary Fig. 2). In control experiments, silencing through local delivery of the GABA-A agonist muscimol had closely comparable impacts on learning to silencing through

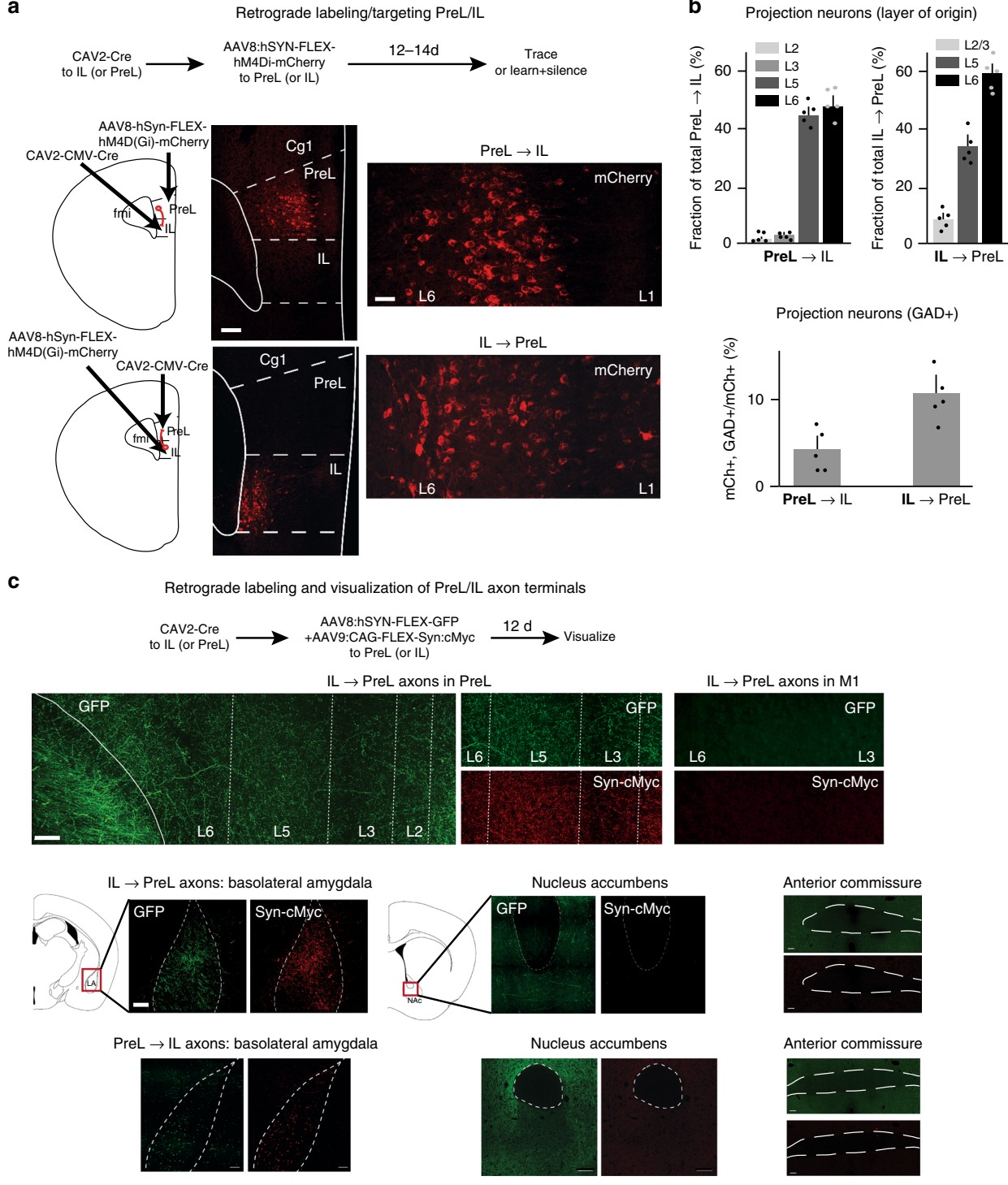

**Fig. 1** Reciprocal PreL/IL layer 5/6 connectivity. **a** Retrograde labeling of PreL→IL and IL→PreL connectivity. Schematic of experimental procedure, and examples of PreL→IL and IL→PreL labeling experiments. Bars: 150 (left), 35 μm (right). **b** Cortical layers from which IL neurons project to PreL or PreL neurons project to IL (in % of total projecting neurons; top), and fraction of GAD + neurons among PreL→IL and IL→PreL projecting neurons (mCh, bottom). $n = 5$ for both PL→IL and IL→PL groups. **c** Analysis of IL→PreL and PreL→IL projecting axon collaterals. Top: schematic of labeling experiments with GFP and synaptophysin-cMyc fusion protein. Panels: examples of IL→PreL and PreL→IL projecting axon (GFP) collateral terminals (Syn-cMyc) in PreL and BLA, but not in M1, nucleus accumbens and anterior commissure. Bar: 300 μm

local PV-neuron activation (Supplementary Fig. 3). Therefore, throughout this study, we consistently silenced PreL or IL through pharmacogenetic PV activation since the silenced region could be verified in every mouse through visualization of confined viral expression with α-Bungarotoxin. Only mice with verified

specific targeting of PreL or IL were included in the analysis (Methods).

We first carried out selective silencing experiments as mice performed subtasks of an IEST[22],[47–49] (Fig. 2b; see Methods). In this sequential learning protocol closely comparable to the

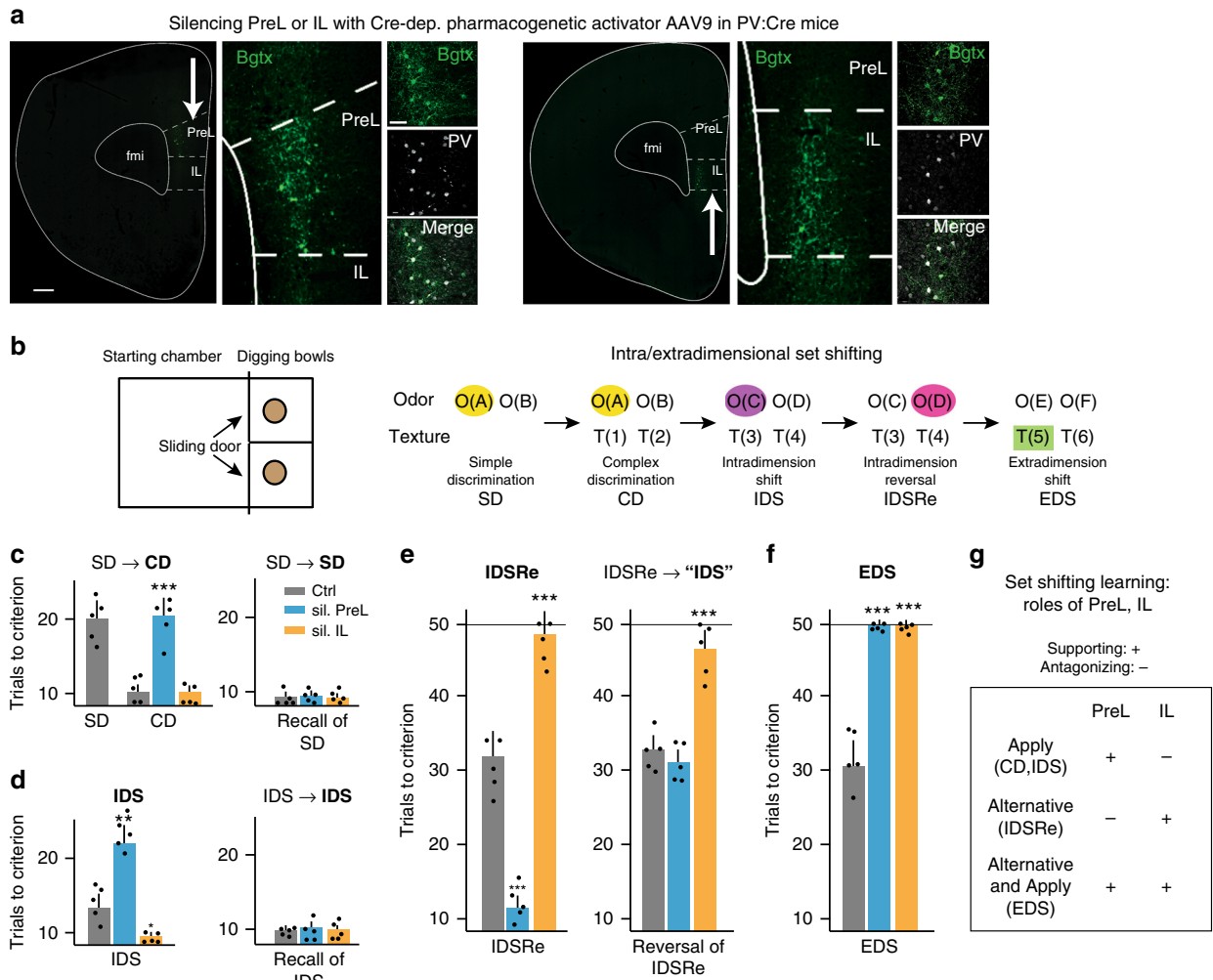

**Fig. 2** PreL is required to apply and IL to shift from learnt rules. **a** Representative images for delivery to PreL or IL of Cre-dependent AAV9-pharmacogenetic activator in *PV-Cre* mice. Bgtx: visualization of (Bungarotoxin + ) virus infected cells; PV: PV immunocytochemistry. Bar: 100 μm (left) and 35 μm (right). **b** Schematic of set-shifting protocol (background color: rewarded association). **c–f** Requirements for activity in PreL or IL during IEST subtask learning. **c** CD ($n = 5$ each; ($F_{(2, 12)} = 17.1$, ***$P$) or recall of SD ($n = 5$ each; $F_{(2, 12)} = 1.9$, $P = 0.1916$ ns); **d** IDS ($n = 5$ each; $F_{(2, 12)} = 24.62$, ***$P$) or recall of IDS ($n = 5$ each; $F_{(2, 12)} = 0.78$, $P = 0.4722$, ns); **e** IDSRe ($n = 5$ each; $F_{(2, 12)} = 233.5$, ***$P$) or reversal of IDSRe ($n = 5$ each; $F_{(2, 12)} = 127.9$, ***$P$); **f** EDS ($n = 5$ each; $F_{(2, 12)} = 43.01$, ***$P$). Bar diagrams: trials to criterion for virus-expressing mice treated with vehicle (Ctrl) or mice in which PreL (blue) or IL (yellow) was silenced (PV-neuron activation; PSEM ligand 20 min before learning); subtask during which silencing was applied indicated in bold. "IDS" indicates that cues are as in IDS, but previous history is distinct from IDS. **g** Summary of PreL and IL roles in IEDS learning. Error bars: SEM; one-way ANOVA followed by Dunnet's post hoc; $P < 0.05$ (*), $0.001$ (**), $0.0001$ (***)

Wisconsin card-sorting task applied in humans to test prefrontal executive functions, mice learn a defined sequence of changing cue/goal associations during five consecutive days. For each of the subtasks, we carried out separate silencing experiments in either PreL or IL by delivering ligand PSEM 20 min before the learning protocol (silencing within 1–2 min, and up to 45–60 min).

On the first day of IEST, mice learned to dig for reward at the cup associated with a particular odor (yellow O(A); simple discrimination, SD; which cup was associated with the rewarded odor changed randomly between the two possible positions), and took about 20 trials to reach criterion (Fig. 2c). On the next day, the odors (O(A) and O(B)) and the particular odor-reward association (yellow O(A)) were left unchanged, but an additional set of (distracting) cues (texture around the cups; T(1) and T(2)) was introduced in the task (complex discrimination, CD). In a first set of experimental mice, PreL or IL was silenced specifically during CD (Fig. 2c). Under the modestly changed circumstances involved in CD, vehicle-treated mice efficiently applied the

previous odor-reward association, now reaching criterion in 9–10 trials (instead of 20 trials on the previous day). In striking contrast, mice with silenced PreL took again about 20 trials to reach criterion (Fig. 2c). Silencing IL did not affect CD performance (Fig. 2c). In a separate set of mice, silencing PreL (or IL) did not affect recall of SD (no distracting textures; Fig. 2c). We, therefore, conclude that mice with silenced PreL failed to apply the previously learned odor-reward association to rapidly learn CD in the presence of potential distractors.

In the next group of experimental mice, PreL or IL was silenced specifically during the next IEST subtask, i.e., intradimensional shift (IDS) (Fig. 2d). In this subtask, new pairs of odors (O(C) and O(D)) and textures (T(3) and T(4)) were introduced, but reward was again associated with one particular odor (blue-violet O(C); Fig. 2b). Vehicle-treated mice rapidly focused on the newly rewarded odor as the correct cue/reward association, and reached criterion in 12–14 trials (Fig. 2d). By contrast, mice with silenced PreL took again 20–22 trials to reach criterion, apparently failing

to apply the previously learned association between the odor dimension and reward to rapidly learn in IDS (Fig. 2d). Mice with silenced IL only needed 9–10 trials to reach criterion in IDS, possibly suggesting an IL-promoted tendency to consider focusing on the alternative (distracting) dimension when none of the two new odors was identical to the previous odor-reward association, thereby slowing IDS learning (Fig. 2d). In a separate group of mice, silencing of PreL or IL did not affect recall of IDS (Fig. 2d), further supporting the notion that activity in PreL and IL is specifically required to promote learning of new cue/reward associations within a given task, but not to retrieve those associations in the absence of variations requiring further learning.

In the next set of experimental mice, PreL or IL was silenced specifically during the subsequent IEST subtask, i.e., intradimensional reversal (IDSRe; Fig. 2e). In this subtask, none of the cues were changed, but mice now had to associate the opposite odor (magenta O(D)) with reward, i.e., they had to learn an alternative to the most recent cue-reward association under conditions identical to those previously associated with the opposite association (Fig. 2b). Strikingly, vehicle-treated mice strongly resisted learning the alternative odor-reward association and required about 30 trials (instead of 20 trials for SD, and 20 trials for CD or IDS with silenced PreL) to reach criterion. Notably, mice with silenced PreL only took 10–11 trials to reach criterion, suggesting that PreL accounted for perseverance with the most recent (now unrewarded) odor-reward association, interfering with learning of the alternative (odor) association (Fig. 2e). In striking contrast, mice with silenced IL mostly failed to reach criterion before 50 trials, suggesting a failure to learn the alternative odor-reward association (Fig. 2e).

To investigate whether activity in IL might be important to learn alternatives to the most recent association, as opposed to being important to learn new associations within the same set of cues, an additional group of mice was presented again with the original IDS cue-reward association after undergoing IDSRe (reversal of IDSRe; Fig. 2e). Vehicle-treated mice took again about 30 trials to reach criterion, suggesting that in the continued presence of identical cues, and in spite of the fact that they had already learned the cue-reward association in IDS before, they resisted overcoming the most recent IDSRe association to learn reversal of IDSRe (Fig. 2e). Mice with silenced IL took again nearly 50 trials to reach criterion, consistent with the notion that activity in IL is important to learn alternatives to the most recent association, regardless of whether these were identical to a previously learned association (Fig. 2e). By contrast, and unlike IDSRe, silencing PreL did not improve performance in reversal of IDSRe (Fig. 2e). That might reflect the fact that the previous IDSRe specifically involved learning an alternative association within the same setting promoted by IL, and not an association learned with the support of PreL.

In a final group of experimental mice, PreL or IL was silenced specifically during the final IEST subtask, i.e., EDS (Fig. 2f). Mice were now presented with two new sets of cues (O(E), O(F), T(5), T(6)), and reward was now associated with a texture (green, T(5)), not an odor. In this case, mice had to learn an alternative dimensional association, and learn to associate a particular texture with reward (Fig. 2b). Vehicle-treated mice apparently resisted shifting from odors to textures as the dimension associated with reward, and took again about 30 trials to reach criterion (Fig. 2f). Notably, and in contrast to the previous IEST subtasks, mice with either silenced PreL or silenced IL both failed to learn EDS within 50 trials (Fig. 2f). These results were consistent with the interpretation that activity in IL is required to learn the alternative dimensional association in EDS (textures instead of odors), whereas activity in PreL

promotes learning of the new association between a texture and reward.

Taken together, these results suggest that when, within the same task, cue-reward associations change and require further learning, activity in PreL is required for application of the previous association rule, thereby supporting (CD, IDS) or interfering (IDSRe) with new learning (Fig. 2g). By contrast, activity in IL is required for learning alternatives to the most recent association, thereby supporting (IDSRe, EDS) or interfering (IDS) with new learning. Finally, activity in PreL is important to again learn a cue-reward association within the new dimension and the same task (EDS, Fig. 2g).

**IL→PreL neurons are required for IL driven shifting in IEST.** We then determined whether and which aspect of reciprocal PreL-IL connectivity might be required in IEST learning. For these experiments, we inhibited IL→PreL or PreL→IL projecting neurons (Fig. 1a) specifically during IDS, IDSRe or EDS learning (total of six different experimental groups of mice; CNO ligand or saline delivered 20 min before learning). Like IL silencing using pharmacogenetic PV-neuron activation, inhibiting IL→PreL projection neurons during learning using inhibitory DREADDs accelerated learning in IDS (Fig. 3a), and suppressed learning in IDSRe (Fig. 3b) and EDS (Fig. 3c). In control experiments, CNO alone or virus alone had no impact on IDSRe learning (Fig. 3b). By contrast, inhibiting PreL→IL projection neurons during learning had no effect on IDS (Fig. 3a), IDSRe (Fig. 3b) or EDS learning (Fig. 3c). Therefore, activity in IL→PreL projection neurons is specifically important during new learning involving the function of IL in IEST learning, whereas activity in PreL→IL projection neurons is not. Furthermore, neither activity in IL→PreL nor in PreL→IL connectivity appeared to be important to mediate the roles of PreL in IEST learning.

To investigate the notion that IL might influence learning by counteracting PreL-mediated application of a previously learned rule, we carried out projection neuron activation experiments during IDS, IDSRe, and EDS learning. This again involved six different groups of experimental mice (CNO or saline 20 min before learning). Indeed, activating IL→PreL projection neurons with activator DREADDs during IDS (Fig. 3d) or IDSRe learning (Fig. 3e) faithfully reproduced the effects of PreL inhibition on set-shifting learning. Again, CNO alone or virus alone had no impact on IDSRe learning (Fig. 3e). Unlike PreL inhibition, activation of IL→PreL projection neurons did not affect performance in EDS learning, consistent with the notion that enhancing the impact of IL on PreL was not sufficient to reproduce the effects of PreL silencing on EDS learning (Fig. 3f). Taken together, these results provided evidence that the direct output of IL to PreL is specifically important for the role of IL during IEST learning.

**PreL promotes fear learning and IL required for its extinction.** To investigate whether relative roles of PreL and IL, and of IL→PreL connectivity in learning might extend beyond IEST, we turned to Pavlovian fear conditioning and its extinction, comparatively simple learning forms in which PreL and IL have been reported to have opposite roles, and where extinction learning could be considered as alternative learning[24,26,30]. We investigated mice that underwent trace fear conditioning (tFC) (Fig. 4a, Supplementary Fig. 4), a form of Pavlovian associative learning involving several brain areas, including medial prefrontal cortex[7,50]. Silencing PreL during acquisition led to absence of freezing at recall 1 day after conditioning, whereas silencing IL during acquisition had no impact on freezing at recall (Fig. 4b). By contrast, neither silencing PreL nor silencing IL at recall

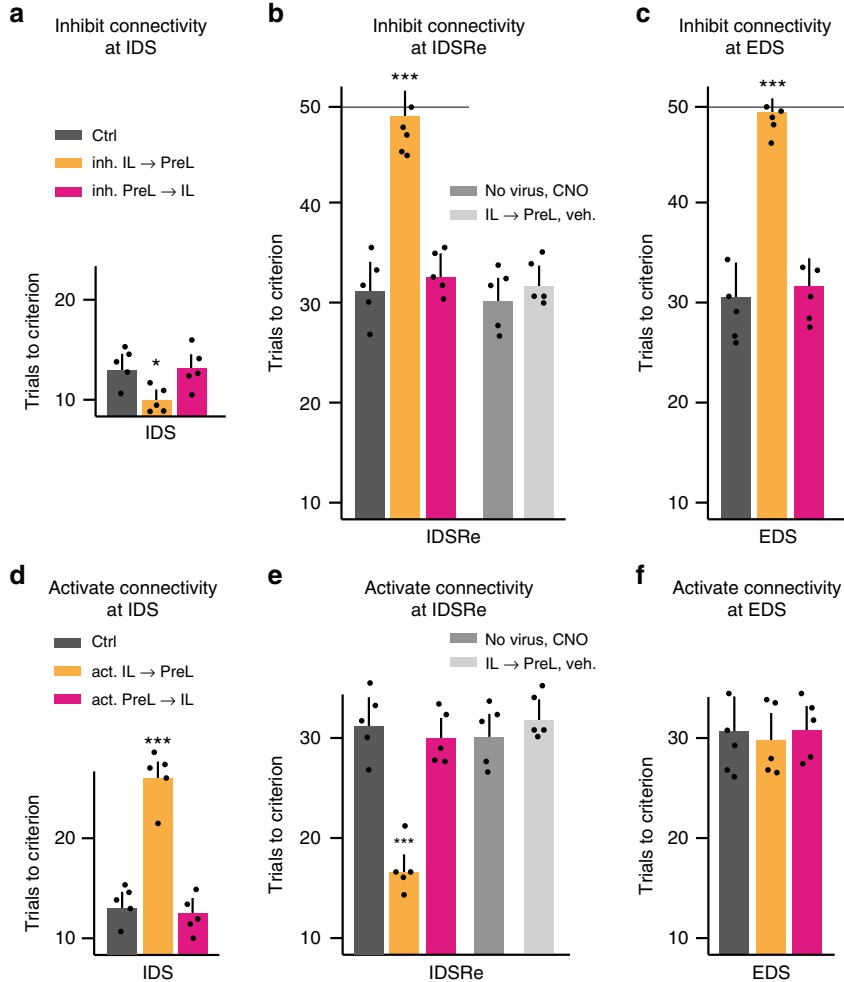

**Fig. 3** IL→PreL neurons are required for IL driven shifting in IEST. **a–c** Silencing IL→PreL, but not PreL→IL projection neurons during IDS (**a**; $n = 5$ each; $F(2, 12) = 6.64$, $^*P = 0.011$), IDSRe (**b**; $n = 5$ each; $F(4, 20) = 17.08$, $^{***}P$), or EDS (**c**; $n = 5$ each; $F(2, 12) = 17.89$, $^{***}P$) learning reproduces the effect of IL silencing during the same IEST subtasks. These experiments involved inhibitory DREADDS into IL or PreL projection neurons, and delivery of CNO or vehicle 20 min before learning; CNO ligand alone or virus alone had no impact on IDSRe learning (**b**). **d, e** Activating IL→PreL, but not PreL→IL projection neurons during IDS (**d**; $n = 5$ each; $F(2, 12) = 25.39$, $^{***}P$) or IDSRe (**e**; $n = 5$ each; $F(4, 20) = 20.01$, $^{***}P$) reproduces the effect of silencing PreL during the same subtasks. **f** Activating IL→PreL projection neurons during EDS has no detectable impact on learning performance ($n = 5$ each; $F(2, 12) = 1.08$, $P = 0.371$, ns). These experiments involved excitatory DREADDS into IL or PreL projection neurons, and delivery of CNO or vehicle 20 min before learning. CNO ligand alone or virus alone had no impact on IDSRe learning (**e**). Error bars: SEM; one-way ANOVA followed by Dunnet's post hoc; $P < 0.05$ ($^*$), 0.001 ($^{**}$), 0.0001 ($^{***}$)

affected freezing during recall (Fig. 4c). In a positive control experiment to verify that interference with recall could be detected with our procedure, silencing ventral hippocampus (vH) at recall effectively suppressed freezing during recall (Fig. 4c)[46]. Therefore, activity in PreL supports learning (but not recall) of tFC, whereas activity in IL is not important at acquisition to learn tFC.

We then investigated requirements for PreL or IL in extinction of tFC. Silencing IL during acquisition of tFC, which did not affect fear learning, also did not affect subsequent extinction of tFC (Fig. 4d). By contrast, silencing IL during the extinction protocol effectively prevented extinction learning (Fig. 4e). Silencing PreL during extinction learning slightly accelerated loss of freezing, while not preventing return of fear 10 d after extinction learning, suggesting that PreL might interfere with extinction learning (Fig. 4e). Finally, silencing IL (or PreL) during recall of extinction learning did not affect absence of freezing during recall of extinction (Fig. 4f).

To further investigate the notion that activity in IL is specifically required for extinction learning, we carried out extinction experiments in which we silenced both, IL required for extinction and PreL that might counteract extinction. Silencing of PreL in addition to IL during the extinction protocol did not noticeably influence how IL silencing interfered with extinction learning (Fig. 4g). In additional experiments, we found undistinguishable dependences on PreL and IL in contextual fear conditioning and its extinction (Supplementary Fig. 5). Taken together, these results provided evidence that activity in PreL has a role in promoting learning of Pavlovian fear conditioning, whereas activity in IL is required for extinction learning in this paradigm (Fig. 4h).

**IL→PreL activity required in IL driven extinction learning**. We next investigated whether, like in IEST, activity in IL→PreL projection neurons might be specifically required during learning

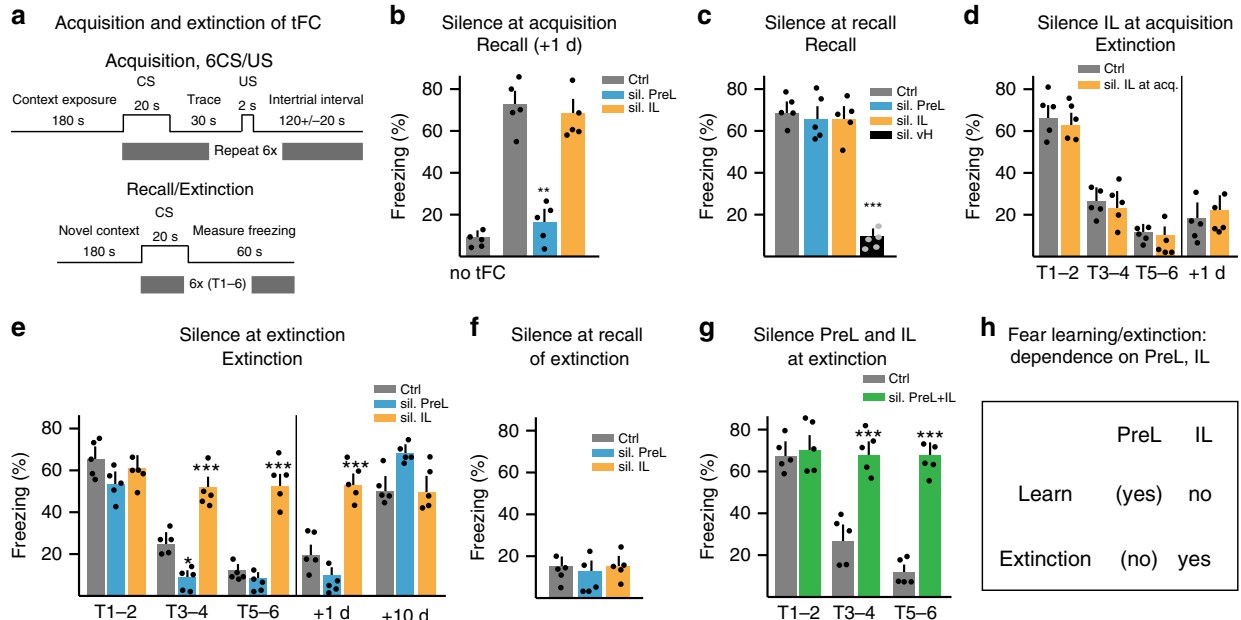

**Fig. 4** PreL promotes fear learning and IL drives its extinction. In all these experiments: silencing through pharmacogenetic activation of PV neurons 20 min before learning (or recall); control: vehicle delivery in virus-expressing mice. **a** Schematic of trace fear conditioning (tFC) and extinction protocol. **b** Silencing PreL at acquisition impairs freezing at 24 h recall, whereas silencing IL does not ($n = 5$ each; one-way ANOVA: $F(2, 12) = 20.67$, ***$P$; Dunnet's post hoc). **c** Silencing PreL or IL at recall does not affect freezing. In positive control experiments, silencing ventral hippocampus (vH) at recall suppresses freezing ($n = 5$ each; one-way ANOVA: $F(3, 16) = 31.14$, ***$P$; Dunnet's post hoc). **d** Silencing IL at acquisition does not affect extinction learning on the next day ($n = 5$ each; repeat measure two-way ANOVA: silence IL, $F(1, 4) = 1.01$, $P = 0.372$, ns). For extinction of tFC, data are shown (here and throughout the paper) as average freezing during T1–T2, T3–T4, or T5–T6; for individual freezing values see Supplementary Fig. 4. **e** Silencing during extinction learning: silencing IL suppresses extinction learning, whereas silencing PreL slightly accelerates loss of freezing ($n = 5$ each; repeat measure two-way ANOVA: silence IL vs. PreL, $F(2, 8) = 64.75$, ***$P$; Tukey's post hoc). **f** Silencing PreL or IL at recall of extinction does not affect recall of extinction learning ($n = 5$ each; one-way ANOVA: $F(2, 12) = 0.636$, $P = 0.546$, ns). **g** Silencing IL and PreL during extinction learning suppresses extinction learning ($n = 5$ each; repeat measure two-way ANOVA: silence PreL + IL, $F(1, 4) = 199.35$, ***$P$; Tukey's post hoc). **h** Summary of PreL and IL roles in fear learning and its extinction. Error bars: SEM. $P < 0.05$ (*), 0.001 (**), 0.0001 (***)

processes that depend on IL in fear and extinction learning. Again, this first involved targeting of specific IL→PreL or PreL→IL connectivity, followed by separate experimental groups of mice, with their controls (CNO ligand or saline delivered 20 min before learning). Inhibiting IL→PreL or PreL→IL projection neurons during acquisition of tFC did not affect freezing to cue-induced recall on the subsequent day (Fig. 5a). By contrast, while inhibiting IL→PreL projection neurons during extinction learning did not affect initial freezing to cue (i.e., recall) it suppressed subsequent extinction learning (Fig. 5b). Furthermore, and in further correspondence to IEST, inhibiting PreL→IL projection neurons during extinction learning did not affect extinction (Fig. 5b).

To investigate whether IL→PreL projecting axons are specifically required for extinction learning through their synaptic terminations in PreL, we silenced their terminals specifically in individual target areas. We expressed a presynaptically targeted silencing DREADDs in IL→PreL (or PreL→IL) projecting neurons, and then delivered ligand (CNO, or saline) locally to specific target areas[51] (Fig. 5c). Silencing the terminals of IL→PreL projecting axons specifically in PreL during extinction learning suppressed extinction learning, whereas silencing of terminations by the same axons in BLA did not suppress extinction learning (Fig. 5c). By contrast, and as predicted, silencing the terminals of PreL-IL projecting axons in IL did not detectably affect extinction learning (Fig. 5c). These results provided evidence that for extinction (but not fear) learning to occur, activity in direct IL→PreL connectivity is specifically required during extinction learning. Furthermore, the combined results from IEST and fear/extinction learning experiments were consistent with the notion

that activity in IL→PreL connectivity is specifically required during learning processes that depend on IL, suggesting that one synaptic output of IL needed for this process is mediated by its projections to PreL.

Studies of extinction after traumatic experiences in humans have highlighted how extinction protocols are not always effective, and that in some cases extinction can be difficult or even not possible to achieve. Accordingly, we wondered whether under circumstances in which extinction might be more challenging to achieve, extinction learning still specifically depends on activity in IL, and on IL→PreL connectivity (Supplementary Notes; Supplementary Figs. 6 and 7). Fear behavior produced by two tFC protocols delivered on consecutive days (2×tFC) resisted subsequent extinction (Supplementary Fig. 6). Consistent with the notion that silencing PreL can accelerate extinction (Fig. 4e), silencing PreL during the extinction protocol of 2×tFC produced robust extinction learning. Silencing of both, PreL and IL during the extinction protocol of 2×tFC again prevented extinction learning, indicating that extinction in the absence of active PreL still depended on active IL. While one extinction session (Ext1) failed to produce detectable extinction of 2×tFC, extinction protocols delivered on three consecutive days (Ext1, Ext2, and Ext3) produced robust extinction upon Ext3 (Supplementary Fig. 6). Notably, silencing IL during Ext1, and Ext2 or Ext3 suppressed detectable extinction learning upon Ext3, indicating that functional IL at both Ext1 and Ext2 was a prerequisite for extinction learning at Ext3. Furthermore, activity in IL→PreL projection neurons was specifically required during processes in Ext1–3 extinction

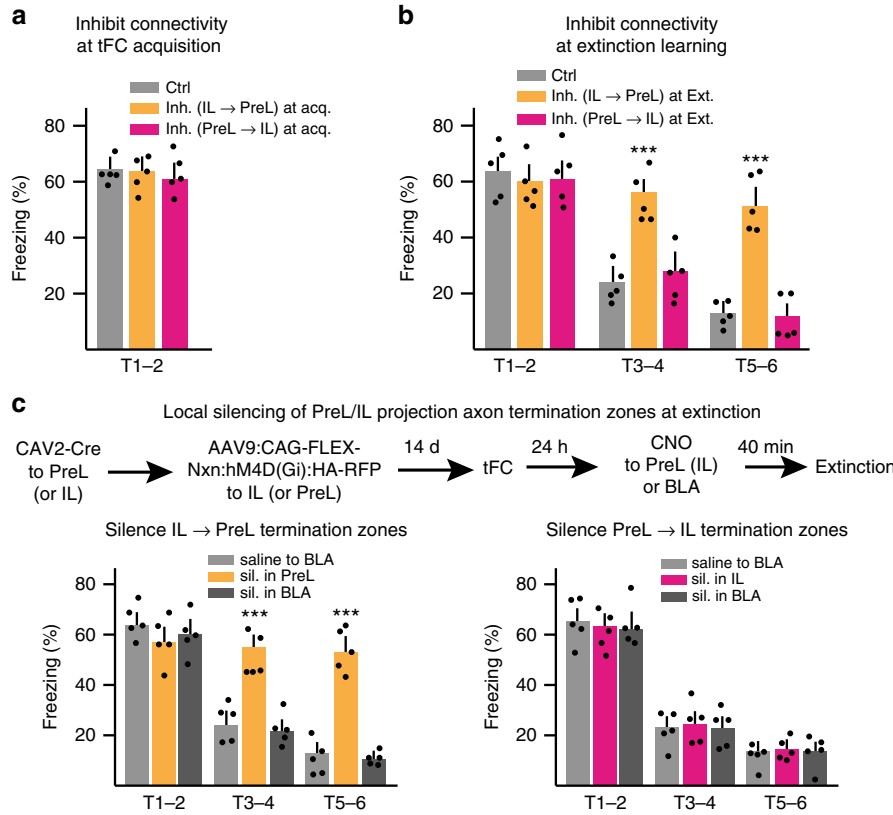

**Fig. 5** IL→PreL activity required in IL driven extinction learning. **a, b** Impact of inhibiting activity in IL→PreL or PreL→IL projection neurons during acquisition of tFC (T1–2 recall values) (**a**; $n = 5$ each; one-way ANOVA: $F(2, 12) = 0.057$, $P = 0.9446$ ns) or during extinction learning (**b**; $n = 5$ each; repeat measure two-way ANOVA: Inh. IL→PreL vs. PreL→IL, $F(2, 8) = 23.22$, ***P). **c** Local silencing of IL→PreL or PreL→IL projecting axon collaterals in their termination zones during extinction learning. Top: schematic of experimental strategy (expression of Neurexin-inhibitory DREADDs fusion protein). Bottom: suppression of extinction learning by silencing of IL→PreL ($n = 5$ each; repeat measure two-way ANOVA: Inh. Terminals in PreL vs. BLA, $F(2, 8) = 58.6$, ***P), but not PreL→IL ($n = 5$ each; repeat measure two-way ANOVA: Inh. Terminals in IL vs. BLA, $F(2, 8) = 2.81$, $P = 0.119$ ns) collateral terminals in PreL (or IL), but not BLA. Error bars: SEM. Tukey's post hoc; $P < 0.05$ (*), $0.001$ (**), $0.0001$ (***)

learning that depend on IL (Supplementary Figs. 6 and 7). Taken together, these results further support the notion that activity in IL→PreL (but not PreL→IL) projection neurons is specifically required during IL-dependent alternative learning.

**IL role in extinction set up 12 h after fear learning in IL.** We then wondered whether there was a time window during which activity in PreL→IL projection neurons might be important in learning involving communication between PreL and IL. As a prerequisite to address this question, we investigated when PreL and IL might undergo learning-related plasticity in fear/extinction learning. We analyzed contents of cFos + neurons in PreL and IL cortical layers upon acquisition, recall and extinction of tFC (Fig. 6a). Ninety min after acquisition of tFC, cFos + neuron contents were greatly increased in PreL layer 2/3, layer 5 and layer 6 (Fig. 6b). By contrast cFos + neuron induction in IL upon acquisition ranged from modest (layer 2/3) to absent (layer 5 and layer 6) (Fig. 6b). Likewise, at recall of tFC (T1–2), cFos + induction was robust in all PreL layers, but either modest (layer 2/3, layer 5) or absent (layer 6) in IL (Fig. 6c). By contrast, and consistent with the notion that IL is specifically involved in extinction learning, extinction of tFC did not produce cFos induction beyond that detected upon recall in PreL, but led to robust induction of cFos expression over recall values in IL (layer 2/3, layer 6, but not layer 5) (Fig. 6d).

We next reasoned that IL might either be specifically recruited for learning at the time of extinction, or that, alternatively,

acquisition of fear learning might already trigger processes in IL important for IL-dependent extinction, and more generally for alternative learning. In a search for processes that might anticipate cFos induction in IL upon extinction learning, we analyzed contents of cFos + neurons in PreL and IL 14 h after acquisition (+14 h), i.e., around the peak of a long-term memory consolidation time window (+12–15 h) known to involve reinduction of cFos expression[46,52]. Notably, while cFos expression at +14 h was not significantly different from that detected upon tFC acquisition in PreL, it was dramatically elevated over acquisition values in IL (Fig. 6e). Specifically, cFos + content values at IL layer 2/3 and layer 6 resembled those induced upon extinction learning, and those in layer 5 dramatically exceeded those detected upon acquisition, recall or extinction learning (Fig. 6e).

To investigate the possible importance of PreL/IL cFos expression at +12–14 h for extinction learning, we delivered a specific small-molecule inhibitor of cFos transcriptional function to either PreL or IL at +12 h[53]. While delivering the cFos inhibitor to PreL did not significantly affect freezing during extinction learning, delivery to IL at +12 h suppressed subsequent extinction learning (Fig. 6f). Therefore, inhibiting cFos from +12 h after acquisition in IL resembled the effect of silencing IL during extinction learning.

To determine how activity in IL or PreL at a +12–14 h time window might influence subsequent learning, we carried out silencing experiments (PV-neuron activation; PSEM delivery at +12 h) in IL or PreL at +12 h. Like inhibition of cFos function,

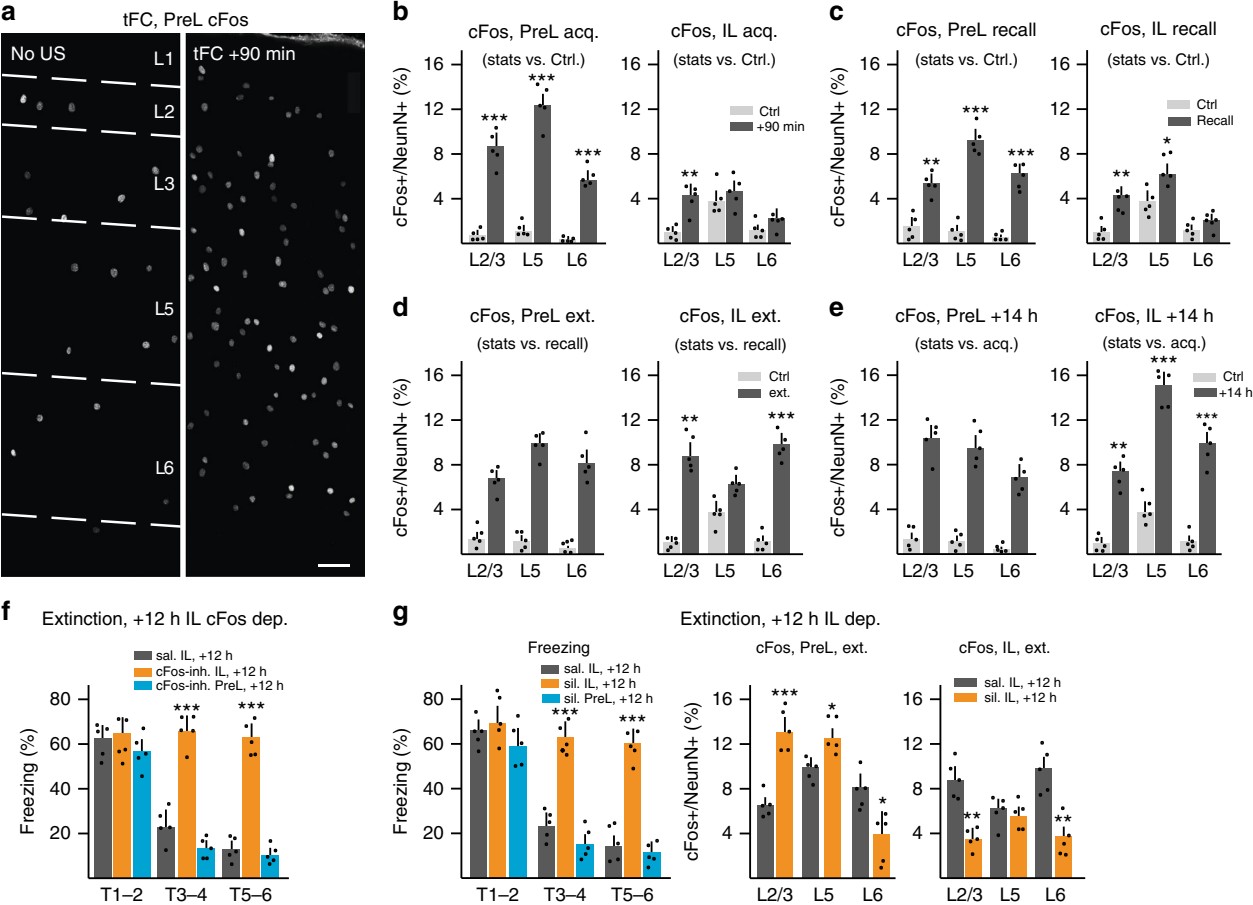

**Fig. 6** Role of IL in extinction is setup 12–14 h postfear learning. **a** Representative images of PreL cFos immunoreactivity upon acquisition in control (CS but no US) or tFC mouse. Bar: 50 μm. **b–e** cFos induction in PreL and IL cortical layers upon acquisition (+90 min; **b**; statistics comparisons to cage control; $n = 5$ each; two-way ANOVA: PreL, $F_{(1, 24)} = 147.5$, ***$P$; IL, $F_{(1, 24)} = 22.4$, ***$P$), upon recall (90 min after T1–2 recall; (**c**) comparisons to cage control; $n = 5$ each; two-way ANOVA: PreL, $F_{(1, 24)} = 163.8$, ***$P$; IL, $F_{(1, 24)} = 25.94$, ***$P$), upon extinction learning (90 min after T1–6; (**d**) comparison to recall; $n = 5$ each; two-way ANOVA: PreL, $F_{(1, 24)} = 2.252$, $P = 0.1465$, ns; IL, $F_{(1, 24)} = 29.57$, ***$P$), or at +14 h after acquisition (**e**) comparison to acquisition; $n = 5$ each; two-way ANOVA: PreL, $F_{(1, 24)} = 0.2994$, $P = 0.5893$, ns; IL, $F_{(1, 24)} = 120$, ***$P$). Sidak's post hoc test for all comparisons in (**b–e**). **f** Effect of cFos inhibition in PreL or IL 12 h after acquisition on next-day extinction learning. ($n = 5$ each; repeat measure two-way ANOVA: cFos Inh. IL vs. PreL, $F_{(2, 8)} = 33.14$, ***$P$; Tukey's post hoc). **g** Effect of PreL or IL silencing 12 h after acquisition on next-day extinction learning (left; $n = 5$ each; repeat measure two-way ANOVA: sil. + 14 h IL vs. PreL, $F_{(2, 8)} = 47.4$, ***$P$; Tukey's post hoc), or on cFos induction 90 min after extinction learning in PreL (center; $n = 5$ each; two-way ANOVA: $F_{(1, 24)} = 8.523$, **$P$; Sidak's post hoc) or in IL (right; $n = 5$ each; two-way ANOVA: $F_{(1, 24)} = 26.79$, ***$P$; Sidak's post hoc). Error bars: SEM; $P < 0.05$ (*), 0.001 (**), 0001 (***)

silencing IL at + 12 h suppressed subsequent extinction learning, whereas silencing PreL did not noticeably affect freezing during extinction learning (Fig. 6g). In parallel, silencing IL at +12 h suppressed cFos induction in IL and led to a marked increase in cFos induction upon the (failing) extinction learning protocol in PreL layer 2/3 (Fig. 6g). Therefore, interfering with cFos or network activity in IL at a +12–14 h time window after fear learning mimics the impact of IL silencing during extinction learning.

**Reciprocal PreL↔IL activity at +12 h for IL function**. We then determined whether activity in IL→PreL and/or in PreL→IL projection neurons 12 h after acquisition might affect subsequent fear memory recall and extinction learning. Inhibiting IL→PreL projection neurons at +12 h after fear learning (DREADDs in IL→PreL neurons, CNO or vehicle at +12 h) did not affect subsequent fear responses to tone (T1–2), but specifically suppressed subsequent extinction of tFC (Fig. 7a). Inhibiting IL→PreL projection neurons at +1 h or at +16 h after fear learning failed to suppress subsequent extinction learning (Fig. 7a), consistent with

the notion that functional interactions between IL and PreL were specifically required during a +12–14 h time window to produce IL-dependent extinction. Notably, inhibiting PreL→IL projection neurons at +12 h after fear learning (DREADDs in PreL→IL neurons, CNO or vehicle at +12 h) also specifically suppressed subsequent extinction of tFC (Fig. 7b), suggesting that IL-dependent extinction learning depended on reciprocal exchange of information between PreL and IL 12–14 h after acquisition of fear learning, much in contrast to the unilateral IL→PreL requirement during extinction learning.

To investigate whether activity in IL→PreL and/or PreL→IL projecting axons is required at +12–14 h for extinction learning specifically through their terminations in PreL (respectively IL), we silenced their terminals specifically in individual target areas (method as in Fig. 5c, but CNO or vehicle locally at +12 h after fear learning). Silencing the terminals of IL→PreL projecting axons in PreL at +12 h suppressed subsequent extinction learning, whereas silencing of terminations by the same axons in BLA did not suppress subsequent extinction learning (Fig. 7c). Likewise, silencing the terminals of PreL-IL projecting axons in IL

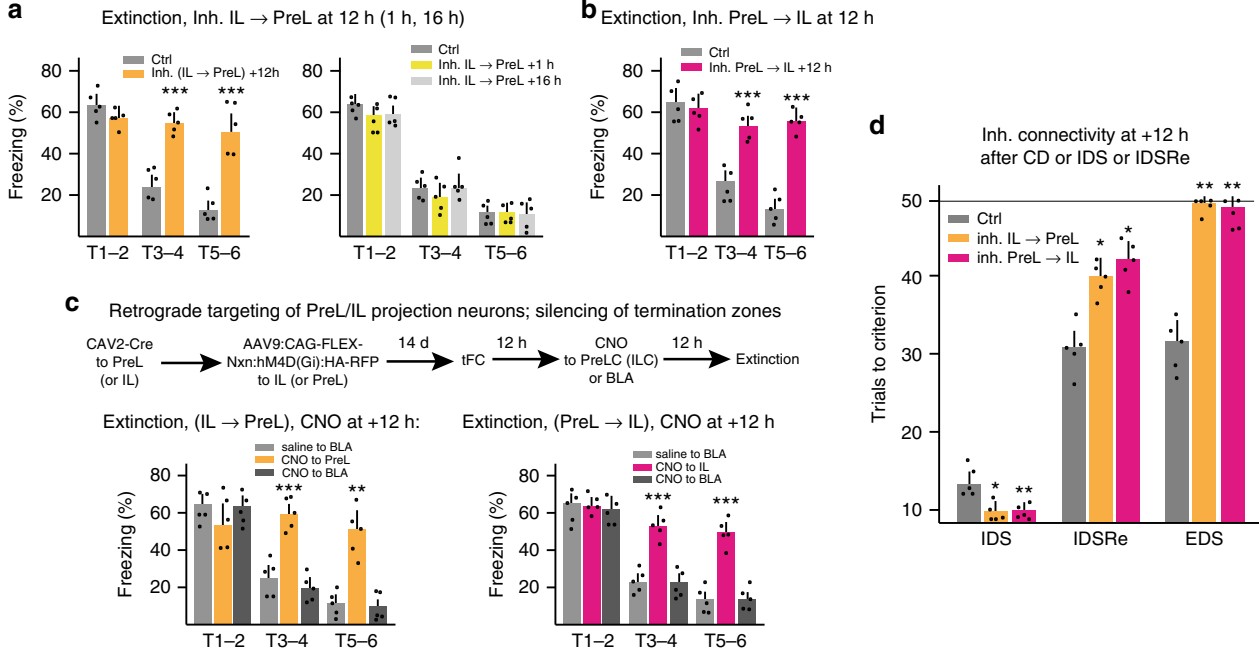

**Fig. 7** Reciprocal PreL↔IL activity at +12–14 h drives IL function. **a, b** Impact of inhibiting IL→PreL (**a**) or PreL→IL (**b**) projection neurons at +12 h after acquisition on subsequent extinction learning. Repeat measure two-way ANOVA; (**a**) $n = 5$ each; Cntrl vs. +12 h, $F_{(1, 4)} = 13.63$, $^{**}P$; (**b**) Cntrl vs. +12 h, $F_{(1, 4)} = 19.21$, $^{**}P$. Silencing IL→PreL at +1 h or +16 h after acquisition did not affect subsequent extinction learning (**a**; $n = 5$ each; repeat measure two-way ANOVA: +1 h vs. +16 h, $F_{(2, 8)} = 1.869$, $P = 0.197$, ns). Sidak's post hoc test for (**a, b**). **c** Local silencing of IL→PreL or PreL→IL projecting axon collaterals in their termination zones at +12 h after acquisition of tFC. Top: schematic of experimental strategy (expression of Neurexin-inhibitory DREADDs fusion protein). Bottom: Suppression of extinction learning by silencing of IL→PreL or PreL→IL collateral terminals in PreL (or IL), but not BLA. Repeat measure two-way ANOVA; $n = 5$ each; IL→PreL: Inh. PreL vs. BLA, $F_{(2, 8)} = 97.73$, $^{***}P$; PreL→IL: Inh. IL vs. BLA, $F_{(2, 8)} = 85.86$, $^{***}P$. Tukey's post hoc test for C. **d** Impact of silencing IL→PreL or PreL→IL projection neurons at +12 h after CD, IDS, or IDSRe learning on subsequent IDS, IDSRe, or EDS learning. Repeat measure two-way ANOVA; $n = 5$ each; Inh. IL→PreL vs. PreL→IL, $F_{(2, 8)} = 36.61$, $^{***}P$; Tukey's post hoc. Error bars: SEM; post hoc; $P < 0.05$ ($^*$), 0.001 ($^{**}$), 0.0001 ($^{***}$)

at +12 h suppressed subsequent extinction learning, whereas silencing them in BLA did not affect subsequent extinction learning (Fig. 7c). Therefore, reciprocal PreL↔IL connectivity is specifically important 12 h after fear learning for subsequent IL-dependent extinction learning.

Consistent with the notion that activity in PreL↔IL reciprocal connectivity at +12 h was specifically required for IL-dependent extinction learning, inhibiting PreL→IL or IL→PreL projection neurons at +12 h led to a complete loss of cFos expression in IL, and to enhanced layer 2/3 cFos expression in PreL upon the extinction protocol (Supplementary Fig. 8). Furthermore, and as might have been expected given the close correspondence between cFos expression at +14 h and upon extinction, inhibiting PreL→IL projection neurons at +12 h produced a complete loss of cFos induction at +14 h in IL (Supplementary Fig. 8). In addition, inhibiting PreL→IL projection neurons at +12 h strongly reduced cFos expression in PreL at +14 h (Supplementary Fig. 8).

Finally, we determined whether activity in PreL↔IL projection neurons 12–14 h after acquisition was also specifically required for the role of IL in set-shifting learning. To address this question we inhibited IL→PreL or PreL→IL at +12 h after CD, IDS, or IDSRe and monitored the impact of these interventions on subsequent IDS, IDSRe, or EDS. As predicted, and closely mimicking the effects of IL silencing during learning, inhibiting IL→PreL or PreL→IL at +12 h after CD improved subsequent IDS learning, whereas inhibiting IL→PreL or PreL→IL at +12 h after IDSRe suppressed subsequent EDS learning (Fig. 7d). The impact of inhibition at +12 h after IDS on IDSRe learning was slightly less pronounced than when IL was silenced during IDSRe learning, but the direction of the change (worsening) was the

same as upon IL silencing (Fig. 7d). Taken together, these results provided evidence that activity in reciprocal PreL↔IL connectivity during a +12–14 h time window after acquisition is essential to set up how IL influences subsequent learning.

## Discussion

We have investigated how two distinct but functionally related prefrontal cortical areas, mouse PreL and IL, exert complementary and possibly opposite roles to support learning. We provide evidence of direct reciprocal PreL↔IL layer 5/6 connectivity, and show that in new learning activity in PreL is important to promote application of previously learned associations, whereas activity in IL is required to learn alternatives to previous associations. Activity in PreL and IL are specifically important when new learning is required, whereas neither activity in PreL nor IL is required for recall of previously learned associations. We further show that activity in IL→PreL connectivity is specifically required during IL-dependent alternative learning, whereas reciprocal PreL↔IL connectivity is specifically required during a time window 12–14 h after learning to set up the role of IL in subsequent alternative learning. Taken together, our results define specific and opposite roles of PreL and IL to flexibly support new learning, and provide circuit-level evidence that direct reciprocal PreL↔IL connectivity is important for activity in IL to mediate learning of associations alternative to those supported by PreL (Supplementary Fig. 9).

The results of our set-shifting experiments suggest that activity in PreL supports new learning by promoting application of relevant previously learned associations. In this way, PreL appears to account for a key aspect of adaptive behavior, allowing to take

advantage of previous related experience to efficiently learn new but related tasks. Our results further show that activity in IL is specifically required to learn alternatives to PreL-promoted associations. Together, PreL and IL, therefore, appear to represent complementary and opposing components of a top-down system to support experience-based flexible goal-oriented learning.

The results of our fear learning and extinction experiments provide further support for the notion that PreL and IL have opposite roles specifically in new learning. Unlike the set-shifting experiments, fear conditioning and its extinction do not directly address rule application and learning of alternative associations under conditions of choice. Accordingly, our results involving such different learning settings cannot be related to one another in every respect. Nevertheless, extinction learning can be considered as learning of an alternative to the CS-US association in Pavlovian fear conditioning, and our results provide evidence that this alternative learning again depends on activity in IL and on IL→PreL connectivity.

Our conclusions are in good agreement with those of a previous lesion study in rats addressing relative roles of PreL and IL in a Y-maze shifting task[21]. Our results are further consistent with previous reports that activity in dorsolateral PFC is correlated to the reward value of action choices[10,16,17], and that medial PFC has critical roles for flexible behavior, particularly strategy shifting[19], in humans, monkeys, and rodents. Previous reports on learning flexibility have focused on the role of IL in promoting flexible behaviors in extinction and addiction[23,29,30,39], and our findings are consistent with the conclusions of those previous studies. Our study does not address how IL is involved in habit learning[4,31]. However, while that likely involves circuits (e.g., prefrontal-striatal) and mechanisms not addressed in this study[33], a possibility consistent with our findings is that context-related shifting away from PreL-promoted goal-directed behavior might favor habitual reward seeking. Taken together, our findings using set shifting and fear/extinction learning protocols lend support to those previous studies that have assigned opposite roles to PreL and IL in learning. By showing how the role of IL in alternative learning depends first on activity in reciprocal PreL→IL and IL→PreL projection neurons 12–14 h after association learning, and then specifically on activity in IL→PreL projections during alternative learning, our results suggest a circuit-level mechanism for how PreL and IL interact to exert opposite influences on learning.

Our finding that activity in IL is required for extinction learning is consistent with the results of previous studies on fear conditioning and its extinction[23,30,39]. Generally, our findings do not support the notion that IL might mediate learning just opposite to that promoted by PreL. This is most evident in EDS learning, which depended on activity in both IL and PreL. Our findings concerning extinction-resistant fear learning suggest that IL-dependent alternative learning is influenced by the strength of previous PreL-dependent learning (see also Ref. [14]). Along similar lines, the results of previous studies involving set-shifting tasks have suggested that if some of the subtasks (e.g., IDSRe) are omitted, EDS can be learned more readily than in the protocol applied in our study, suggesting that the extent of previous reinforcement is important in set-shifting[49,54]. Taken together, these considerations suggest a conceptual framework in which application and perseverance with previously learned associations is balanced against alternation and shifting in learning. Our data suggest that this balance process is influenced by the combined outcome of previous learning experiences, possibly including the relative strengths of PreL- and IL-dependent memories. However, how sequences of learning sessions are integrated to influence subsequent learning, and the extent to

which PreL, IL, and other prefrontal areas might be involved remain to be determined.

A key result of our study is that PreL and IL exhibit direct reciprocal layer 5/6 connectivity, and that this connectivity is critically important for IL-dependent alternative learning. The specific requirement for activity in IL→PreL, but not PreL→IL connectivity during IL-dependent learning suggests that PreL has an important role in IL-dependent learning, and that PreL and IL might have antagonistic roles in controlling flexible learning. Consistent with this notion, inhibition of IL→PreL connectivity prevented IL-dependent learning, whereas activation of IL→PreL connectivity during IEST learning mimicked PreL silencing. How IL→PreL connectivity might be recruited to interfere with PreL-mediated perseverance and promote alternative learning remains to be determined. Furthermore, our results make no specific predictions as to how connectivity to and from downstream brain areas mediates the distinct effects of activity in PreL and IL on learning.

The reciprocal connectivity between PreL and IL mainly involving layer 5/6 can potentially complicate the interpretation of our interference experiments. Thus, for example, influencing activity in IL→PreL connectivity might, in turn, influence activity in PreL→IL connectivity, making it difficult to disentangle specific roles of these individual circuitries in learning. Our study does not address how activity in PreL→IL or in IL→PreL projection neurons influences local circuit activity in their respective target areas. However, the behavioral outcomes of our manipulations provide evidence for striking functional asymmetries in the impact of PreL→IL and of IL→PreL connectivity on learning. Thus, inhibiting IL→PreL connectivity activity by either specifically inhibiting the projection neurons themselves, or by inhibiting activity in the corresponding collaterals specifically in their target area in PreL closely mimicked the effects of inhibiting activity in IL as a whole (thereby presumably suppressing all outputs from IL). Furthermore, enhancing activity in IL→PreL projection neurons mimicked effects of inhibiting activity in PreL as a whole, suggesting that activity in IL→PreL connectivity is critically and specifically important to support IL-promoted alternative learning, as opposed to PreL-promoted rule application. By contrast, inhibiting activity in PreL-IL connectivity not only did not mimic the effects of inhibiting PreL (i.e., we found no symmetry to the outcome of the IL→PreL interference experiments), but it specifically interfered with subsequent IL-dependent alternative learning when carried out during a memory consolidation time window 12–14 h after acquisition. These findings are consistent with a conceptual framework in which rule learning occurs first (PreL-dependent), and the role of IL in learning alternatives to that rule is set up subsequently, in a process that involves activity in PreL→IL projection neurons. Our findings are further consistent with the possibility that activity in IL→PreL projection neurons might control subpopulations of neurons in PreL with a specific role in steering IL-dependent learning. One possible model consistent with our findings could involve projections from PreL and IL to shared downstream areas in which behaviorally important learning processes might occur in a PreL and/or IL-directed manner, and in which the corresponding memories would be stored for behavioral recall. In such a model, some of the projections from PreL to downstream areas might interfere with IL-dependent alternative learning, and the interference might be suppressed by IL→PreL connectivity. This arrangement would be consistent with our finding that inhibiting activity in PreL or IL had no detectable behavioral impact during recall of previous learning. The hypothetical model does not imply that activity in IL→PreL connectivity is the only route through which PreL and IL influence each other's impacts in new learning, and additional systems, e.g., involving cortico-striatal-

thalamic circuitry might also be involved. In sum, our results provide evidence for specific roles of PreL→IL and IL→PreL connectivity to support IL-promoted learning of alternatives to PreL-promoted associations. Elucidating the precise circuit mechanisms and the broader systems arrangements through which PreL and IL influence rule application vs. shifting in flexible learning will, however, require further research.

A further key result of our study is that the role of activity in IL to promote learning of alternative associations is set up during a time window 12–14 h after acquisition of the original association. Such 12–14 h time windows are thought to be important for long-term memory consolidation through processes involving coordinated activity between distributed networks[55,56]. Information exchange between PreL and IL might therefore occur in the context of such systems-wide processes[57], and the direct layer 5/6 connectivity between PreL and IL might not be the only route through which balances between PreL and IL memories are established to modulate the outcome of rule application vs. alternative associations in learning[39,58–60]. The late time frame for setting up systems balances that define subsequent learning might ensure that relevant information obtained before or after the initial acquisition can be included into an updated representation of learned rules and their values. The mechanisms through which activity at +12–14 h is specifically required to establish PreL/IL balances that regulate persistence vs. alternative learning remain to be determined.

In more general terms, our results highlight unique roles of late memory consolidation processes at +12–14 h in flexible learning. Our results suggest that balancing the relative importance of future conflicting strategies might require integration processes best achieved during off-line memory consolidation processes.

## Methods

**Mice.** *PV-Cre* mice from Jackson laboratories (*129P2-Pvalbtm1 (cre)Arbr/J*) were a kind gift from S. Arber (Friedrich Miescher Institut). Mice were kept in temperature-controlled rooms on a constant 12 h light-dark cycle, and all experiments were conducted at approximately the same time of the light cycle. Before the behavioral experiment, mice were housed individually for 3–4 d and provided with food and water *ad libitum* unless otherwise stated. All animal procedures were approved and performed in accordance with the Veterinary Department of the Kanton Basel-Stadt.

**Behavioral procedures.** All experiments were carried out with male mice. At the onset of the experiments P60 mice were 55–65 d old.

For trace fear conditioning (tFC)[50], animals were placed in the training context (Habitest Unit, Coulbourn Instruments, Allentown, PA) and were given a 180 s habituation period. Conditioning trials began with a 20 s mixed-frequency tone (CS), followed by a 30 s trace period before the animal received a 2 s (0.8 mA) foot shock (US). The conditioning trial was followed by a pseudo-random intertrial interval (ITI) of 120 ± 20 s. The conditioning trials were repeated 6 times to get a robust CS–US association. To assess retrieval of trace fear memory the animals were placed in a novel context (a white circular enclosure in a separate noise proof cabinet) and given a 180 s habituation period. Retrieval trials began by presentation of a 20 s tone and freezing was measured for 60 s posttone presentation, in the absence of foot shock. In extinction experiments this was repeated six times with a variable ITI of 90 ± 30 s postfreezing measurement to get six independent measures of freezing and a gradual extinction curve. Identities of the contexts were maintained with the presence of distinct odors; 2% acetic acid for training context and 0.25% benzaldehyde for retrieval context. The training and retrieval chambers were cleaned with 70% ethanol before and after each session. Control mice were subjected to the same procedure without receiving foot shocks. Freezing was defined as complete absence of somatic mobility other than respiratory movements.

Contextual fear conditioning experiments were carried out as described[61]. Briefly, mice were allowed to explore the training context for 2.5 min, and then received five foot shocks (1 s and 0.8 mA each, intertrial interval: 30 s). Control mice were subjected to the same procedure without receiving foot shocks. We assessed contextual fear memory by returning mice to the training chamber after fear conditioning during 5 min, and analyzing freezing during a test period of 4 min (first min excluded). In extinction experiments mice were kept in the training chamber for 30 min without foot shocks.

For IEST, mice were trained to dig in plastic food bowls, with an internal diameter of 40 mm and a depth of 40 mm and filled with wood chips, to retrieve a food reward buried at the bottom of the bowl. The reward was one-third of a

Honey Nut Loop (Kellogg, Manchester, UK). The outer surface of the bowl was covered with a texture and the rim of the bowl was coated with an odor. The test apparatus was a rectangular Plexiglas box with panels that divided one-third of its length into two sections of equal proportions. The digging bowls were placed in these sections and two doors with independent access separated the mouse from the two sections where the bowls were placed. The doors could be slid open to give the mouse access to the food bowls or closed rapidly to prevent access to the other bowl after an error. The mice were food restricted from a day before the habituation phase till the end of the experiment. Each day, after the end of testing, mice were given access to 1 g of a mash made of crushed food pellets mixed with water. The habituation phase consisted of 3 days; each day mice were given access to 2 bowls filled with wood chips and baited with reward. Mice were allowed to dig in both the bowls for the reward following which the bowls were rebaited for a second time. During test phases, trials were initiated by opening both doors simultaneously to give the mouse access to the two digging bowls, only one of which was baited. The first four trials were exploratory trials, where the mouse was allowed to dig in both the bowls and an error was recorded if the mouse dug first in the unbaited bowl. On subsequent trials access to the other bowl was occluded as soon as the mouse began digging in one of the bowls. A correct trial was recorded when the mouse dug in the baited bowl to collect the reward. A trial was terminated when the mouse dug in the correct bowl and collected the reward, or dug in the un-baited bowl and left the chamber. Testing continued until the mouse reached a criterion level defined as 8 correct trials out of the last 10 trials. IEST consisted of five testing sessions spread over 5 days, each involving a different type of discrimination. In simple discrimination the bowls differed along the odor dimension only. For compound discrimination a second dimension (textures) was introduced, but correct and incorrect odors remained constant. For IDS a new pair of odors and textures were introduced and the mouse had to learn a new odor-reward association using the odor rule learnt previously. For IDS reversal the odor and texture cues remained unchanged but the mouse had to learn that the previously correct odor-reward association was now incorrect and the previously incorrect odor now predicted reward. Finally, for the extradimensional shift a new pair of odors and textures were introduced and the previously relevant dimension (odor) was now irrelevant whereas the previously irrelevant dimension (texture) now predicted reward. The order of the discriminations was always the same, but the dimensions and the pairs of cues were equally represented within groups and counterbalanced between groups.

**Immunocytochemistry and histology.** Antibodies used were as follows: goat anti-PV (Swant biotechnologies, PVG-213) 1:5000; rabbit anti-cFos (Santa Cruz biotechnology, sc-52) 1:8000; mouse anti-NeuN (Millipore, MAB377) 1:1000; α-Bungarotoxin, Alexa488 Conjugate (Molecular Probes, Life Technologies, B-13422) 1:500; mouse α-cMyc (ATCC, CRL-1729) 1:1000 and mouse α-GAD67 (Abcam; ab26116) 1:500. Secondary antibodies were Alexa Fluor 488 (Molecular Probes; A150077), or 647 (Molecular Probes; A150131, A150107); 1:500. Prelimbic and Infralimbic were at +1.8 to +1.95 mm from bregma while Basolateral Amygdala and Nuclear Accumbens were at +1.8 and +0.8 mm from bregma, respectively. Samples belonging to the same experiment (for example, from the mice of a given time point, with their controls) were acquired in parallel and with the same settings on an LSM700 confocal microscope (Zeiss) using an EC Plan-Neofluar340/1.3 oil-immersion objective (Zeiss). cFos immunohistochemistry and intensity analysis were done as described[46]. For cFos analysis, mice were returned to their home cage for 90 min or till the relevant time point after the training session and then perfused (transcardially with 4% PFA in PBS, pH 7.4). Brains were kept in fixation solution overnight at 4 °C and cryostat sections were processed for immunocytochemistry. All cFos and/or NeuN immunopositive cells were quantified using an automatic spot-detection algorithm (Imaris 7.0.0, Bitplane AG; expected radius, 10 mm; quality level, 7). cFos induction was quantified as a fraction of cFos positive cells over the total neuronal population expressing NeuN.

In experiments involving injections with picospritzer or Hamilton needles, brains were collected at the end of the experiments for histological analysis (e.g., Figure 2A). Serial slices were imaged at 10× or at 4× to locate the injection site and the extent of volume spread. All viral expression spread were controlled with α-Bungarotoxin staining for PSAM channel or mCherrry signal for the DREADDS. Occasional mice in which the injections had also targeted neighboring areas were excluded from the analysis.

**Stereotaxic surgery.** All surgeries were conducted under aseptic conditions using a small animal stereotaxic instrument (David Kopf Instruments). Mice were anaesthetized with isoflurane (4% for induction, 1.5–2.0% afterward) in the stereotaxic frame for the entire surgery and body temperature was maintained with a heating pad. Local drug treatments were carried out with a 33-gauge needle coupled to a 5ul syringe (Hamilton, Reno, NV), while viruses were delivered using glass pipettes (tip diameter 10–20 μm) connected to a picospritzer (Parker Hannifin Corporation). Coordinates relative to bregma are as follows: PreL (anteroposterior (AP) + 2.0 mm, mediolateral (ML) + 0.5 mm, dorsoventral (DV, relative to dura) −2.1 mm); IL (angle (ML–DV plane; away from mid-line) 10°, AP + 1.8 mm, ML + 1.0 mm, DV −2.95 mm); vH (AP −3.0 mm, ML + 2.9 mm, DV −3.5 mm) and BLA (AP −1.6 mm, ML + 3.0 mm, DV −4.3 mm). For drug injections the needle was slowly lowered to 0.1 mm beyond the required DV

coordinate and quickly pulled up to the original coordinate to create a pocket for injection of the drug. This prevented backflow of the drug and undesirable spread into neighboring areas. Drugs were injected at the rate of 100 nl/min to a final maximum volume of ~200 nl. After completion of injection the needle was left in its place for 5 min to allow for diffusion of the drug and then slowly withdrawn. For virus injections the glass pipette was inserted at the desired coordinate and ~200 nl of the virus preparation was slowly pressure injected using a picospritzer over a period of 10 min. After the end of the injection the pipette was left in its place for a further 10 min to allow for diffusion of the virus. All drugs and viruses were injected bilaterally in the PreL, IL, vH, and BLA. Postsurgical recovery was monitored daily until the start of behavioral protocols. All injections were paired with saline injected control animals to account for any effect due to tissue damage or surgical procedure.

**Pharmacology in vivo**. Drugs were used bilaterally as follows: Muscimol (50 ng/side prepared in buffered saline, Sigma); T-5224 (1ug/side prepared in 20% PVP and 10% DMSO, MedChemExpress; cFos-AP1 inhibitor[53]).

**Pharmacogenetic in vivo**. For acute silencing of a brain area, we bilaterally delivered floxed PSAM-carrying AAV9 (excitation, rAAV9-CAG-flox-PSAM (Leu41Phe,Tyr115Phe)5HT3-WPRE) in *PV-Cre* mice[45,46,61]. To allow for transgene expression, mice were kept under home cage conditions for 7–9 d before any behavioral experiment. The PSAM agonist, PSEM308 was injected i.p. at 5 mg/kg of animal weight to activate the PSAM channels[45]. All silencing experiments were done 20 mins before the behavior.

For PreL→IL or IL→PreL activation/silencing experiments a canine adenovirus 2 (CAV2) carrying Cre-recombinase (IGMM, France) was injected bilaterally in the PreL (or IL). Concurrently, a FLEXed DREADD-carrying AAV8 (Sternson and Roth, 2014) (B. Roth, UNC Vector Core; activation, rAAV8-hSyn-DIO-hM3D (Gq)-mCherry; silencing, rAAV8-AAV-hSyn-DIO-hM4D(Gi)-mCherry) was injected in the IL (or PreL). To visualize collaterals and terminals of IL→PreL or PreL→IL projections, beyond the prefrontal cortex, a similar approach as above was taken. Here instead of DREADDs a FLEXed Synaptophysin-cMyc carrying AAV9 (a kind gift from S. Arber, FMI) and a FLEXed GFP carrying AAV8 (B. Roth, UNC Vector Core) was coinjected into the IL (or PreL). Thus we could follow the projection fibers using the GFP signal and their synaptic terminals by immunostaining for cMyc. For chemogenetic synaptic silencing of the projection neurons a FLEXed synaptically targeted silencing DREADD[51] was used. To allow for transgene expression, mice were kept under home cage conditions for 12–14 days before any behavioral experiment. Selective activation of the DREADD receptors was achieved by administering the DREADD agonist, clozapine-N-oxide (CNO; 5 mg/kg, i.p, Tocris)[41]. For synaptic silencing CNO (0.3 μM) was administered through a Hamilton needle at target regions. All DREADD activation experiments were done 30 mins before the behavioral procedure. Post experiment brains were dissected and serial slices were imaged at 10× to verify correct projection neuron labeling using the mCherry signal.

**Statistical analysis**. All statistical analyses were based on two-tailed comparisons and were done using GraphPad Prism (GraphPad software. Inc.). Results are presented as mean ± s.e.m. The sample size per group and statistical tests are mentioned in the respective figure legends. Number of animals to be used for a standard behavioral analysis was determined based on our preliminary behavioral experiments across different investigators in the laboratory. We paid particular attention to the magnitude of difference between groups tested apart from the statistical significance across groups. Data distributions were assumed to be normal but this was not formally tested. Male mice of closely comparable age were assigned randomly to experimental groups. Intensity analysis and freezing data were verified by a co-worker blind to experimental conditions.

**Data availability**. All relevant data files are available from the authors upon request.

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

## Acknowledgments

We thank S. Arber (FMI) for valuable comments on the manuscript. This work was supported in part by a Swiss National Fund grant to P.C. and by the NCCR Synapsy. The Friedrich Miescher Institut is part of the Novartis Research Foundation. This project has received funding from the European Research Council (ERC) under the European Union's Horizon 2020 research and innovation programme (MemoryDynamics - grant agreement No 694086).

## Author contributions

A.M. devised and carried out all the experiments. P.C. helped devise the experiments and wrote the manuscript. All authors discussed the results and commented the manuscript.
