## [Peer Review File · Nature Communications]

Reviewers' comments:

Reviewer #1 (Remarks to the Author):

In this manuscript, "Infralimbic cortex required to learn alternatives to prelimbic-promoted associations through reciprocal connectivity." Mukherjee and Caroni observe that there are reciprocal connections between layers 5 and 6 of the infralimbic (IL) and prelimbic (PL) regions of the mouse medial prefrontal cortex. Then they use several different chemical genetic methods to manipulate activity in these regions during different phases of a battery of set shifting tasks, and during conditioned fear learning and extinction. There are a number of interesting observations here, and some themes emerge from the data that confirm and enrich existing ideas about the relationship between these two regions and their distinct relationships to learning. Specifically, the authors find results that are broadly consistent with the push and pull of IL and PL with regard to fear memory establishment and extinction. However, they expand that concept into a broader framework where establishing initial associations and applying them to modified contexts seems to be the major role of the PL. Conversely, IL seems to be more important for updating associations previously established by the PL. Unfortunately in my opinion the manuscript suffers from two major issues. First, there are so many experiments, and they are so densely presented, that the result is very confusing and hard to synthesize. I would suggest to the authors that these data might be better served if split up into more than one paper. Second, many of the experiments involve selective inactivation of neurons in one region that project to the other region. This reciprocal connectivity makes it very difficult to interpret the results because manipulation of one region is not dissociable from manipulation of the other region.

Below I elaborate on these concerns.

1) In general as a reviewer, I try to limit my role to evaluating the quality of the science, and not to insert myself into stylistic matters. However, in this case I really feel that some of the choices the authors make about the scope of the study and which data to include present serious clarity issues. The sheer number of permutations in the experiments necessitate a very densely written manuscript that I felt was very hard to follow. I leave it up to the authors and the editor to decide how best to handle that, but one suggestion might be to remove the analysis of the time-dependency of the requirement for IL->PL neurons (the experiments in which these neurons are inactivated in the period between fear conditioning and extinction.) Another possibility is to remove the variations in the conditioning protocol that lead to IL-dependent or IL-and PL-dependent extinction. To my mind, these are interesting side roads, but are not central to the primary findings that are highlighted in the title and abstract. In that sense, they may be seen as a distraction.

2) I respect the fact that the authors are trying to take our understanding of the distinct roles of the IL and the PL to a deeper level of circuitry, but the mutual entanglement of these regions at the circuit level work against that goal and complicate interpretation of their experiments. At first in figure 2, the authors manipulate the two regions separately during their battery of set shifting tasks, and they find their most compelling observations. These are that PL is important for applying established associations, and that IL is important for revising those associations. This is a nice extension of some of what was already known about these two regions, and it casts previous results in a new more general light. Things get more complicated when the authors selectively perturb a population in IL (or PL) that gives rise to a projection to PL (or IL.) This manipulation can be expected to actually perturb activity in both regions. For example, if one manipulates IL->PL neurons, those neurons will perturb activity in PL through their projections to PL, but they may also make collaterals within IL itself. Moreover, the resulting altered activity in PL, might be expected to indirectly perturb IL by altering the activity of the reciprocal connectivity from PL back to IL. Fundamentally, it is not clear whether

manipulating IL->PL is a manipulation of IL or PL. The answer is, it is both. The difficulties presented by this become clear when one tries to connect the results of this connectivity manipulation with the results of the regional manipulation of IL alone or PL alone. For example, in figure 4E the authors show that silencing IL prevents extinction of fear memory, but silencing PL does not. Then in figure 5B, the authors show that inhibiting IL-> PL prevents extinction of fear memory, but inhibiting PL-> IL does not. At first it might seem like a sensible result, however it's clear that something more complicated is afoot, because that result means that PL circuitry is required for fear extinction. The terminals of IL->PL would obviously be disabled in experiments that inactivate all of PL, yet those experiments do not impair fear extinction. I suspect there is some more complex dynamic relationship or balance between the activity in each of these regions plays an important role in different types of learning, and these experiments here don't really reveal that yet. As a result, the data seem inconclusive to me.

Reviewer #2 (Remarks to the Author):

This is an interesting manuscript describing the discovery of reciprocal connections between the PL and IL and the potential functional role of these reciprocally connected neurons in learning. The data are novel and in general the methods are state of the art and sound. Below are concerns with the presentation that makes the paper difficult for a reader to track, and concerns with over-interpretation.

1. The description of Figure 1 seems incomplete. I get that space restrictions may be hampering a complete description, in which case a more complete description should be delegated to the Supplement. Specifically, the authors refer to amygdala collaterals but not accumbens collaterals from IL to PL neurons. They did the converse experiment with PL to IL collaterals, but there is not description or micrographs from this experiment. Moreover, as the authors probably know, the projections of the IL and PL to the ventral striatum are quite topographically organized, and the negative data shown (micrograph 1c) does not give confidence that the authors fully evaluated the IL axon terminal fields in the accumbens to allow the negative conclusion made.
2. While the 5HT3 ligand appears to be a nice approach, the authors should briefly comment on the lack of sensitivity to endogenous 5HT since this transgene is not as recognized as the more standard use of DREADDs.
3. This paper is very difficult to follow and the results required multiple readings. It would be helpful if the authors included a small panel outlining the experimental design and even the conclusions if possible. This would allow the reader to quickly track this complex set of behavioral experiments, even if they are not so familiar with the tests employed.
4. Throughout the figure legends, the authors need to specifically state the N in each group. This could be in the legend text or put into each bar. While the data seem statistically robust, the N is an important factor for the reader to evaluate the relative power of the studies, especially for the various negative data.
5. The conclusion that neither the PL or IL is involved in recall seems to contradict the work with Pavlovian cues in the addiction literature where PL inactivation, in particular the projection to the accumbens is required for expression of cue-induced drug-seeking behavior.
6. The last sentence of the last complete paragraph on pg 18 doesn't make sense.

7. A caveat that the authors should consider including is that while they find influences by neurons projecting between the PL and IL, which is the primary finding of interest in this report, the studies of necessary roles do not discount the possibility that other circuits play a role. In particular, interactions between the IL and PL could involve a broader cortico-striato-thalamic circuit which contains a multisynapse mechanism for producing the same thing as the direct connection between IL to PL. Effects of the PL to IL project are not so easily replicated by the known multisynapse circuitry. Along these lines the authors might check out EurJNeuro 35, 614, 2012

Reviewer #3 (Remarks to the Author):

A quite extraordinary amount of work has been packed into this manuscript by Mukherjee and Caroni. The focus is on the prelimbic (PL) and infralimbic (IL) cortex of the mouse. IL and PL are already known, from a number of studies, to have complementary, even opposite, effects on learning and decision making but so far a mechanistic understanding of why this is the case has been lacking. In a very impressive series of studies the authors show: 1) that there are connections between PL and IL; 2) they are focused on layers 5 and 6; 3) that the effects of PL and IL silencing on a learning and set switching task can be mimicked by various combinations of inhibition and activation of the connections between IL and PL; 4) analogous effects can be found in a distinct behavioural task: fear conditioning.

The worst that can be said of the manuscript is that the rather dense style that is needed to convey so much information makes it a tough read at several points. This is particularly the case when the results shown in figure 6 are described. Perhaps this is because the figure 6 results begin to describe a rather new set of ideas regarding critical periods for IL and PreL activity during fear learning rather than just focussing on PreL / IL interactions as in the previous sections. Nevertheless the results are clear and should be of interest to a wide audience. I have just a few minor points to draw the authors' attention to.

Minor points

1 Line 62 It is argued that rodent infralimbic (IL) and prelimbic (PL) resemble primate dorsolateral prefrontal cortex. I understand the point that the authors are trying to make. However, others argue that rodent IL and PL areas resemble the IL and PL areas that can be identified in primates (Wise 2008, Vogt 2009). Maybe it would be fairer to say that there are broad similarities in aspects of frontal cortical function in rodents and primates and that the frontal areas present in rodents, such as IL and PL, appear to be important for flexible behaviour just as, broadly speaking, prefrontal cortex is in primates?

2 Figure 3c. Why does inhibition of PreL-IL projecting neuron does not have a disruptive effect on extra-dimensional set shifting (EDS)? PL silencing affected EDS in figure 2. In other words, inhibiting the projections from PreL to IL does not have the same effect as simply silencing PreL. However, as the authors note the complementary manipulation, the IL-PreL projection neuron manipulation, exerts effects that resemble those of the IL lesions in figure 2.

3 P9, Figure 3 Given that neither inhibition or activation of PreL-IL had any effect on any of the task elements measured in figure 3 can the authors be sure that the experimental manipulation was effective?

A similar argument might be made in relation to the results presented in figure 6 in the fear learning paradigms. Manipulating PreL-IL connections in figure 6g did not disrupt a behaviour that had been shown to be PreL dependent in figure 6d.

Did any aspect of behaviour change as a result of this manipulation? At the very least a few words on the potential limitations of the approach and the inferences that can be made with it would be useful.

4 Line 131-2 Why were slightly oblique orientations used when attempting to silence PreL or IL: "Specific targeting of PreL or IL was achieved under stereotactic guidance through slightly oblique orientation delivery angle"?

5 Figure 2b. The colours and shapes that are used in this figure help draw the reader's attention to the correspondence between this mouse task and the classic Wisconsin card sorting task but at the expense of understanding the nature of the texture and odour discriminations that are actually tested. Maybe the figure should emphasize the odour and texture discriminations?

Typos

Line 51 "...by keeping this information into working memory.." should be "...by keeping this information in working memory..."

Line 551-552 Should "Accordingly, our results involving such different learning settings cannot be related to another in every respect." be "Accordingly, our results involving such different learning settings cannot be related to one another in every respect."

Vogt, B. (2009). Architecture, neurocytology, and comparative organization of monkey and human cingulate cortices. Cingulate Neurobiology and Disease. B. Vogt. New York, Oxford University Press.

Wise, S. P. (2008). "Forward frontal fields: phylogeny and fundamental function." Trends Neurosci 31(12): 599-608.

We thank the three Reviewers for their supportive comments on our manuscript NCOMMS-17-30324, and for their valuable and constructive suggestions for improvements. We have acted upon all the comments, and our detailed responses are listed below in blue (Reviewer comments in black; corresponding changes in the text of the manuscript in red).

We believe that the manuscript could be substantially further strengthened as a result of these revisions, and hope that the Reviewers will now find it suited for publication in *Nature Communications*.

Reviewer #1 (Remarks to the Author):

There are a number of interesting observations here, and some themes emerge from the data that confirm and enrich existing ideas about the relationship between these two regions and their distinct relationships to learning. Specifically, the authors find results that are broadly consistent with the push and pull of IL and PL with regard to fear memory establishment and extinction. However, they expand that concept into a broader framework where establishing initial associations and applying them to modified contexts seems to be the major role of the PL. Conversely, IL seems to be more important for updating associations previously established by the PL.

Thank you for this assessment of the work's advance.

Unfortunately in my opinion the manuscript suffers from two major issues. First, there are so many experiments, and they are so densely presented, that the result is very confusing and hard to synthesize. I would suggest to the authors that these data might be better served if split up into more than one paper. Second, many of the experiments involve selective inactivation of neurons in one region that project to the other region. This reciprocal connectivity makes it very difficult to interpret the results because manipulation of one region is not dissociable from manipulation of the other region.

Below I elaborate on these concerns.

1) In general as a reviewer, I try to limit my role to evaluating the quality of the science, and not to insert myself into stylistic matters. However, in this case I really feel that some of the choices the authors make about the scope of the study and which data to include present serious clarity issues. The sheer number of permutations in the experiments necessitates a very densely written manuscript that I felt was very hard to follow.

We thank the Reviewer for providing valuable suggestions on how to improve on the clarity of the paper.

I leave it up to the authors and the editor to decide how best to handle that, but one suggestion might be to remove the analysis of the time-dependency of the requirement for IL->PL neurons (the experiments in which these neurons are inactivated in the period between fear conditioning and extinction.)

We considered this first option. In our opinion, what would argue in favor of removing the part on time-dependency is that we currently can't provide a circuit account of what happens during the 12-14h time window after acquisition that modifies the PreL-IL network in such a way that subsequent shifting learning becomes possible.

On the other hand, what argues in favor of keeping the part on time-dependency is that the experiments provide a setting for when activity in PreL->IL connectivity has a critical role for subsequent learning. Notably, the results show that the role of activity in PreL->IL connectivity does not mirror that of IL->PreL connectivity (one conceivable outcome could have been that PreL and IL have symmetrical influences on each other's outputs: with other words, IL->PreL would support IL roles (which it does), whereas PreL->IL would support PreL roles (which it does not)). In addition, without the data, readers might be left wondering as to whether PreL->IL is ever behaviorally important (and also, whether we might see no effects because we somehow fail to have sufficient impact on activity in PreL->IL connectivity in our experiments).

For these reasons, and because we think that the alternative suggestion by the Reviewer (below) also results in message simplification while having a smaller impact on the main results of the study, we opted for the alternative suggestion. A further reason to opt for the alternative is that Reviewer 3 makes a very similar suggestion.

Another possibility is to remove the variations in the conditioning protocol that lead to IL-dependent or IL-and PL-dependent extinction. To my mind, these are interesting side roads, but are not central to the primary findings that are highlighted in the title and abstract. In that sense, they may be seen as a distraction.

We agree with this assessment and have now moved these findings to a new supplementary figure (Suppl. Fig. 6). In results we now only provide a brief summary of these experiments, pointing out that they provide further evidence that shifting is consistently IL-dependent. The original, much more detailed text is now in Suppl. Material. This change should substantially simplify the part of the paper that deals with fear learning and its extinction. We could also completely remove these data from the paper, if the Reviewer thinks that it would be best (we would suggest keeping them because the findings should be of interest to the fear learning/extinction field).

2) I respect the fact that the authors are trying to take our understanding of the distinct roles of the IL and the PL to a deeper level of circuitry, but the mutual entanglement of these regions at the circuit level work against that goal and complicate interpretation of their experiments. At first in figure 2, the authors manipulate the two regions separately during their battery of set shifting tasks, and they find their most compelling observations. These are that PL is important for applying established associations, and that IL is important for revising those associations. This is a nice extension of some of what was already known about these two regions, and it casts previous results in a new more general light.

Thank you for these remarks.

Things get more complicated when the authors selectively perturb a population in IL (or PL) that gives rise to a projection to PL (or IL.) This manipulation can be expected to actually perturb activity in both regions. For example, if one manipulates IL->PL neurons, those neurons will perturb activity in PL through their projections to PL, but they may also make collaterals within IL itself. Moreover, the resulting altered activity in PL, might be expected to indirectly perturb IL by altering the activity of the reciprocal connectivity from PL back to IL. Fundamentally, it is not clear whether manipulating IL->PL is a manipulation of IL or PL. The answer is, it is both.

We completely agree with the Reviewer that manipulating PreL->IL or IL->PreL projection neurons is ultimately a perturbation of both, PreL and IL. Generally, we agree with the Reviewer's assessment, and agree that these aspects have the potential to substantially complicate the interpretation of the data involving manipulation of activity in IL->PreL and PreL->IL connectivity. We take this important point, and now bring up these issues explicitly in the Discussion section (second paragraph of second subsection (new)).

We point out that we address some of the potential confounds by including experiments in which we inhibit the outputs of the connections in specific target areas (e.g. local inhibition in PreL of IL->PreL connectivity output) instead of inhibiting the neurons of origin and hence all collaterals made by these projection neurons (Figs. 5c, 7c). Further experiments addressing the interpretation of these manipulations involve showing that activation (instead of inhibition) of IL->PreL connectivity enhances IL-dependent learning (Fig. 3e), and interferes with PreL-dependent learning (Fig. 3d). Taken together, our results suggest that although activity in IL->PreL connectivity targets neurons in PreL (and indirectly possibly also in IL) and hence ultimately the output of PreL, at the behavioral level activity in IL->PreL connectivity preferentially affects IL-dependent learning. This finding suggests that activity in IL->PreL might target a specific subset of neurons in PreL, and that these neurons might have specific roles in steering IL-dependent learning.

We further point out in the Discussion that while our manipulations could have had several different possible outcomes, we in fact find that (at the behavioral level) the reciprocal connectivity (i.e. IL->PreL and PreL->IL connectivity) is particularly important for how IL promotes alternative learning. We think that the striking asymmetry of how activity in PreL->IL and in IL->PreL connectivity affects learning (both preferentially promoting IL-dependent learning) is a key result in our study (now highlighted in the Discussion). That, in our opinion, might be consistent with the notion that learning and consolidating a rule might precede learning alternatives (updates) to that rule (now mentioned in the Discussion) – indeed, we detect little cFos induction in IL upon rule acquisition or its recall, and robust cFos induction in IL at 15h after acquisition, and upon alternative learning (here extinction) (Fig. 6).

Based on these observations, we also suggest one possible arrangement consistent with our findings: PreL and IL might project to shared downstream areas in which behaviorally important learning might occur (in a PreL and/or IL-directed manner), and the corresponding memory might be stored. In this model, some of the projections from PreL might interfere with IL-dependent alternative learning. This arrangement

might also account for the finding that inhibiting activity in PreL or IL had no detectable behavioral impact during recall of previous learning.

The difficulties presented by this become clear when one tries to connect the results of this connectivity manipulation with the results of the regional manipulation of IL alone or PL alone. For example, in figure 4E the authors show that silencing IL prevents extinction of fear memory, but silencing PL does not. Then in figure 5B, the authors show that inhibiting IL-> PL prevents extinction of fear memory, but inhibiting PL-> IL does not. At first it might seem like a sensible result, however it's clear that something more complicated is afoot, because that result means that PL circuitry is required for fear extinction. The terminals of IL->PL would obviously be disabled in experiments that inactivate all of PL, yet those experiments do not impair fear extinction. I suspect there is some more complex dynamic relationship or balance between the activity in each of these regions plays an important role in different types of learning, and these experiments here don't really reveal that yet. As a result, the data seem inconclusive to me.

Again, we generally agree with the Reviewer that the local circuit processes underlying these findings are likely complex.

We now argue (see new paragraph in the Discussion, and see also above) that when inhibiting activity in the whole IL, we suppress IL outputs essential for alternative learning, and suggest that when inhibiting activity in IL->PreL connectivity (by either inhibiting the projection neurons or their outputs specifically in PreL), we might suppress IL-directed activity in PreL that can disinhibit shifting. (Incidentally, for clarification, inhibiting activity in the whole of PreL through PV neuron activation should suppress all outputs of PreL and in this sense (as the Reviewer implies) also suppress any impact of IL->PreL connectivity onto the output of PreL neurons, although strictly speaking the manipulation (PV neuron activation in PreL) should not disable the actual terminals of IL->PreL projection axons). Consistent with these ideas, inhibition of PreL greatly accelerated alternative learning in IDSR_e (Fig. 2e, left; providing evidence that activity in PreL can interfere with alternative learning; see also PreL-prevented extinction), and activation of IL->PreL connectivity enhanced IL-dependent learning (Fig. 3e). A further key finding in this context is that suppressing activity in PreL+IL had the same effect as suppressing activity in only IL (Fig. 4g), or as inhibiting activity in IL->PreL connectivity (i.e. alternative learning is suppressed), whereas suppressing activity in only PreL did not interfere with (and in fact facilitated) alternative learning.

Taken together, these results suggest that the output of IL is essential for alternative learning, and that one output of PreL (functionally inhibited by IL->PreL connectivity) might interfere with IL-dependent alternative learning. We now bring up these considerations in the Discussion.

In sum, we agree with the Reviewer that much remains to be elucidated concerning how reciprocal IL<->PreL connectivity affects rule application versus shifting, but believe that our findings do provide evidence that the output of IL is essential for alternative learning, and that (at the behavioral level) IL->PreL (and PreL->IL) connectivity is specifically important for how IL manages to promote alternative learning.

We now bring up these considerations in the Discussion, and at the same occasion point out how the identification of the specific circuit mechanisms will require further studies.

Reviewer #2 (Remarks to the Author):

This is an interesting manuscript describing the discovery of reciprocal connections between the PL and IL and the potential functional role of these reciprocally connected neurons in learning. The data are novel and in general the methods are state of the art and sound.

Thank you for this positive assessment of our study.

Below are concerns with the presentation that makes the paper difficult for a reader to track, and concerns with over-interpretation.

1. The description of Figure 1 seems incomplete. I get that space restrictions may be hampering a complete description, in which case a more complete description should be delegated to the Supplement. Specifically, the authors refer to amygdala collaterals but not accumbens collaterals from IL to PL neurons. They did the converse experiment with PL to IL collaterals, but there is not description or micrographs from this experiment. Moreover, as the authors probably know, the projections of the IL and PL to the ventral striatum are quite topographically organized, and the negative data shown (micrograph 1c) does not give confidence that the authors fully evaluated the IL axon terminal fields in the accumbens to allow the negative conclusion made.

We now added panels for PreL to IL collaterals, as suggested. Again, we see collaterals with presynaptic terminals in BLA but not in ventral striatum. We re-examined the collateral data (for IL->PreL and for PreL->IL projection neurons) in many sections of NAcc and could not find synaptophysin+ terminals (although we did detect GFP+ axons). We now mention in Results that although we failed to detect synaptophysin+ collateral terminals in NAcc, we cannot exclude the possibility that projections targeting circumscribed areas within NAcc might have gone unnoticed in our analysis.

It should be noted that in these experiments we are only looking at collaterals of IL->PreL projection neurons or PreL->IL projection neurons that also target the NAcc. These will likely be a subset of all neurons that project out of PreL or IL. So it is possible that this particular subset of PreL or IL projection neurons does not target the NAcc, and the majority of PreL or IL projections seen in NAcc belong to a different subset of IL and/or PreL derived projection neurons.

2. While the 5HT3 ligand appears to be a nice approach, the authors should briefly comment on the lack of sensitivity to endogenous 5HT since this transgene is not as recognized as the more standard use of DREADDs.

We now added the information concerning lack of responsiveness to endogenous 5HT to the Results section, as suggested. Briefly, the ion pore domain taken from the 5HT3 receptor only works in conjugation with the ligand-binding domain, which is a modified alpha7 muscarinic ACh receptor (therefore, endogenous 5HT or ACh cannot

functionally activate the PSAM construct). Furthermore, and as shown in the original Sternson paper, the artificial ligand PSEM exhibits poor binding to endogenous alpha7 muscarinic ACh receptor or to endogenous 5HT3 receptor.

3. This paper is very difficult to follow and the results required multiple readings. It would be helpful if the authors included a small panel outlining the experimental design and even the conclusions if possible. This would allow the reader to quickly track this complex set of behavioral experiments, even if they are not so familiar with the tests employed.

We thank the Reviewer for this suggestion. We now added a supplementary figure (Suppl. Fig. 9) with a schematic representation of the main behavioral experiments and their outcome, and refer to it in the Discussion section.

Furthermore, as suggested by Reviewers 1 and 3, to reduce the complexity of the paper we removed the findings on PreL-prevented extinction (previous Fig. 6) from the main results.

4. Throughout the figure legends, the authors need to specifically state the N in each group. This could be in the legend text or put into each bar. While the data seem statistically robust, the N is an important factor for the reader to evaluate the relative power of the studies, especially for the various negative data.

We now included in the figure legends the N values for each group of data, as requested.

5. The conclusion that neither the PL nor IL is involved in recall seems to contradict the work with Pavlovian cues in the addiction literature where PL inactivation, in particular the projection to the accumbens is required for expression of cue-induced drug-seeking behavior.

We tested the effects of PreL or IL silencing in several different recall settings throughout this study, and could not detect any behavioral consequence of the silencing during recall. An interpretation consistent with these findings (and with some previous models of PreL/IL function) could be that PreL and IL direct new learning in downstream structures, which are then essential for recall. We now mention this possibility in the Discussion.

We have not investigated drug-seeking behavior and therefore cannot comment based on our data. The apparent contradiction might reflect different experimental approaches, or possibly a different circuit logic in the drug seeking learning experiments. We could mention this specifically in the Discussion if the Reviewer thinks that it would be important (we now do mention alternative circuits, as suggested by the Reviewer in point 7).

6. The last sentence of the last complete paragraph on pg 18 doesn't make sense.

We take the Reviewer's point, and removed the last sentence from the discussion section.

7. A caveat that the authors should consider including is that while they find influences by neurons projecting between the PL and IL, which is the primary finding

of interest in this report, the studies of necessary roles do not discount the possibility that other circuits play a role. In particular, interactions between the IL and PL could involve a broader cortico-striato-thalamic circuit which contains a multisynapse mechanism for producing the same thing as the direct connection between IL to PL. Effects of the PL to IL project are not so easily replicated by the known multisynapse circuitry. Along these lines the authors might check out EurJNeuro 35, 614, 2012

We agree that showing that PreL \leftrightarrow IL projections are important for alternative learning does not imply that other projections and systems are not important. We take this important point by the Reviewer, and now bring up this caveat in the Discussion, where we also mention cortico-striato-thalamic circuits as a possibility (end of second paragraph of last section). One further possibility could be that PreL/IL (or in some cases possibly either of those two mPFC areas alone) might be involved in several types of learning processes that utilize in part distinct systems, and that our findings apply to some but not all flexible learning settings.

Reviewer #3 (Remarks to the Author):

A quite extraordinary amount of work has been packed into this manuscript by Mukherjee and Caroni. The focus is on the prelimbic (PL) and infralimbic (IL) cortex of the mouse. IL and PL are already known, from a number of studies, to have complementary, even opposite, effects on learning and decision making but so far a mechanistic understanding of why this is the case has been lacking. In a very impressive series of studies the authors show: 1) that there are connections between PL and IL; 2) they are focused on layers 5 and 6; 3) that the effects of PL and IL silencing on a learning and set switching task can be mimicked by various combinations of inhibition and activation of the connections between IL and PL; 4) analogous effects can be found in a distinct behavioural task: fear conditioning.

Thank you for this kind assessment of our findings.

The worst that can be said of the manuscript is that the rather dense style that is needed to convey so much information makes it a tough read at several points. This is particularly the case when the results shown in figure 6 are described. Perhaps this is because the figure 6 results begin to describe a rather new set of ideas regarding critical periods for IL and PreL activity during fear learning rather than just focussing on PreL / IL interactions as in the previous sections.

We agree with the Reviewer's point. In response to these comments and to similar ones by Reviewer 1, we have moved the data in Fig.6 to a new Suppl. Fig. 6, moved most of the corresponding text (Results) to Supplementary material, and only left a brief synthesis of the main point (i.e. that even when extinction is much more challenging to achieve, it remains a learning process depending on activity in IL and in IL \rightarrow PreL connectivity, and that activity in PreL can interfere with alternative learning (here extinction)). We also removed treatment of these findings from the Discussion. We agree that these findings were not absolutely essential to our main points, and hope that this will significantly facilitate reading of the manuscript.

Nevertheless the results are clear and should be of interest to a wide audience.

Thank you.

I have just a few minor points to draw the authors' attention to.

Minor points

1 Line 62 It is argued that rodent infralimbic (IL) and prelimbic (PL) resemble primate dorsolateral prefrontal cortex. I understand the point that the authors are trying to make. However, others argue that rodent IL and PL areas resemble the IL and PL areas that can be identified in primates (Wise 2008, Vogt 2009). Maybe it would be fairer to say that there are broad similarities in aspects of frontal cortical function in rodents and primates and that the frontal areas present in rodents, such as IL and PL, appear to be important for flexible behaviour just as, broadly speaking, prefrontal cortex is in primates?

We thank the Reviewer for this suggestion. We now modified the Introduction text accordingly. When referring to dorsolateral PFC we now say "a prefrontal area related by some studies to rodent medial PFC". Furthermore, after summarizing putative roles for rodent PreL and IL in flexible learning, we now added the sentence: "Comparable important roles for flexible behavior as those assigned to rodent PreL and IL, have been assigned to primate PFC."

2 Figure 3c. Why does inhibition of PreL-IL projecting neuron does not have a disruptive effect on extra-dimensional set shifting (EDS)? PL silencing affected EDS in figure 2. In other words, inhibiting the projections from PreL to IL does not have the same effect as simply silencing PreL. However, as the authors note the complementary manipulation, the IL-PreL projection neuron manipulation, exerts effects that resemble those of the IL lesions in figure 2.

The Reviewer makes an important point. We find that inhibiting PreL->IL connectivity 12h after IDSRe learning interferes with subsequent EDS learning (Fig. 7d), whereas inhibition of PreL->IL connectivity during EDS learning had no detectable behavioral impact (Fig. 3c). These findings are closely comparable to how inhibiting PreL->IL connectivity at +12h after fear learning interfered with subsequent extinction learning (Fig. 7b), whereas inhibition during extinction learning did not affect extinction learning (Fig. 5b).

More generally, these results show that during new learning the two types of directional connections (IL->PreL and PreL->IL) do not have a symmetrical impact on learning (this could have been the case, but we find that it is not; as the Reviewer points out, inhibiting IL as a whole (i.e. presumably all of its outputs) had closely comparable consequences during new learning as specifically inhibiting IL->PreL connectivity, whereas, by contrast, inhibiting PreL->IL connectivity during new learning had no detectable effect, and inhibiting PreL prevented rule application in new learning). With other words, IL->PreL and PreL->IL connectivity are both predominantly important for IL-dependent alternative learning. We don't have a conclusive answer for how this works at the level of local circuit activity, but think that the finding might be easiest to explain if PreL and IL might direct new learning at one or more downstream areas. That would be consistent with the notion that these prefrontal areas are important during new learning but not for its recall (some of the

downstream areas would be critical for new learning and for its retrieval; one example we noticed is hippocampal circuitry). One possibility could then be that IL->PreL interferes with critical activity in PreL that might counteract the impact of IL on downstream circuitry essential for shifting (in a sense dis-inhibition of shifting). By contrast, PreL->IL might be important to induce plasticity in IL essential for subsequent control of shifting by IL->PreL connectivity. Such a model would be consistent with push-pull ideas. Conceptually, our findings seem to us consistent with the notion that PreL has an "upstream role" in shifting because IL controls shifting from PreL-dependent rules. We now mention some of these ideas in the Discussion.

3 P9, Figure 3 Given that neither inhibition or activation of PreL-IL had any effect on any of the task elements measured in figure 3 can the authors be sure that the experimental manipulation was effective?

A similar argument might be made in relation to the results presented in figure 6 in the fear learning paradigms. Manipulating PreL-IL connections in figure 6g did not disrupt a behaviour that had been shown to be PreL dependent in figure 6d.

Did any aspect of behaviour change as a result of this manipulation? At the very least a few words on the potential limitations of the approach and the inferences that can be made with it would be useful.

Inhibiting PreL->IL connectivity 12h after learning had closely comparable consequences on subsequent shifting as inhibiting IL->PreL (Fig. 7) or IL itself (Fig. 6g). We therefore have no reason to assume that the manipulation itself was not effective when applied during alternative learning (when interfering with activity in IL->PreL had effects closely comparable to those of inhibiting IL itself, but inhibiting PreL->IL connectivity was ineffective).

With respect to comparisons between the impact of PreL silencing and silencing of PreL->IL connectivity, we think that while PreL silencing interferes with all outputs of PreL (not just those to IL), inhibition of PreL->IL connectivity only affect one specific subset of PreL outputs. This is consistent with the observation that inhibiting PreL->IL only had detectable behavioral consequences in our experiments when carried out 12h after the original rule learning, whereas silencing of PreL during new learning did have a behavioral impact in several of our experiments. We now bring up these considerations in the Discussion, and at the same occasion point out how the identification of the specific circuit mechanisms involved will require further studies.

4 Line 131-2 Why were slightly oblique orientations used when attempting to silence PreL or IL: "Specific targeting of PreL or IL was achieved under stereotactic guidance through slightly oblique orientation delivery angle"?

We now mention the reason in Results, and reformulate the statement to specify that the oblique angle was important to selectively target IL: we used an oblique angle in order to reach IL while leaving PreL untouched. This further ensured that drugs and viruses remained confined to PreL and IL. The angle (10 degree angle to the midline) was derived empirically through error pilot experiments, and allowed access to IL without "touching" PreL (the needle goes through the fimbria fornix).

5 Figure 2b. The colours and shapes that are used in this figure help draw the reader's attention to the correspondence between this mouse task and the classic Wisconsin

card sorting task but at the expense of understanding the nature of the texture and odour discriminations that are actually tested. Maybe the figure should emphasize the odour and texture discriminations?

We agree that the schematic in Fig. 2b was not sufficiently effective in directing the reader's attention to the key aspects of the task, and thank the Reviewer for this valuable suggestion. We now modified the schematic in Fig. 2b accordingly. We hope that it now better helps readers to focus on the key odor and text discriminations in the task. It is possible that the Reviewer in fact meant to specifically mention which odor and texture are being used - if so, we will be happy to modify again the figure.

Typos

Line 51 "...by keeping this information into working memory.." should be "...by keeping this information in working memory..."

Thank you, now fixed.

Line 551-552 Should "Accordingly, our results involving such different learning settings cannot be related to another in every respect." be "Accordingly, our results involving such different learning settings cannot be related to one another in every respect."

Thank you, now fixed.

Vogt, B. (2009). Architecture, neurocytology, and comparative organization of monkey and human cingulate cortices. *Cingulate Neurobiology and Disease*. B. Vogt. New York, Oxford University Press.

Wise, S. P. (2008). "Forward frontal fields: phylogeny and fundamental function." *Trends Neurosci* 31(12): 599-608.

REVIEWERS' COMMENTS:

Reviewer #1 (Remarks to the Author):

First, like the other reviewers, I found the first submission to be very dense and hard to read. Some measure of this was due to the large number of experiments, and I therefore suggested removing some of the experiments from the Results and main figures. I suggested two possibilities for experiments to be removed (actually they were moved to Supplementary, but the effect is similar), and the authors opted for one of those options. Having now re-read the manuscript, I agree with the choice they made. I agree that the section they removed was the more dispensable one, and that the dataset they kept was more essential. The result is, in my opinion, more readable and flows better. The manuscript is still very demanding, but I think some of that is inherent to the study due to its complexity.

Second, I had some reservations about the interpretability of the experiments targeting specific cell populations that form reciprocal pathways between PL and IL. It seems that at least one other reviewer shared this general concern. The authors have largely addressed these reservations with a substantial new passage in the discussion. In it, they acknowledge the broad issue, highlight aspects of their data that seem to favor a conclusion that these reciprocal pathways are functionally asymmetric, and lay out several specific scenarios for interactivity between the pathways. For example, the authors suggest that the PL->IL and IL->PL pathways might interact through collaterals that converge in overlapping downstream targets. This is an interesting, but speculative possibility. It's probably fair to say that the authors and I don't completely agree about the relative likelihood of contributions from these specific scenarios, but they represent a broad class of interpretive caveats that are important to include. In that sense, the authors have provided a thoughtful consideration of these caveats that improves the manuscript.

One final note: The final two sentences on page 18 that summarize their results with regard to fear conditioning behavior are kind of a mouthful. I had to really work to parse them. The authors draw some important conclusions here, so I recommend for the sake of clarity that they break the section up into more digestible pieces.

Editorial Note: Reviewer 2 sent his/her comments confidentially, which were paraphrased for the authors.

Reviewer #3 (Remarks to the Author):

The authors have addressed all the comments I made in a reasonable and balanced manner. I continue to think that this is an unusually impressive study and recommend it for publication.

Editorial Note: The authors have paraphrased the remaining actionable concerns of the reviewers (in blue), including those of Reviewer 2, which we provided to them in our decision letter.

Answers to Reviewer's comments:

Reviewer1:

Suggests modifying the last two sentences on Page 18 for clarity.

Previous version: Taken together, these considerations suggest a conceptual framework in which how application and perseverance with previously learned associations is balanced against alternation and shifting in learning is influenced by the combined outcome of previous learning experiences, possibly including the relative strengths of PreL- and IL-dependent memories. However, how sequences of learning sessions are integrated to influence subsequent learning, and the extent to which PreL, IL as well as further prefrontal areas such as anterior cingulate and orbitofrontal cortex might be involved remains to be determined.

New version: Taken together, these considerations suggest a conceptual framework in which application and perseverance with previously learned associations is balanced against alternation and shifting in learning. Our data suggest that this balance process is influenced by the combined outcome of previous learning experiences, possibly including the relative strengths of PreL- and IL-dependent memories. However, how sequences of learning sessions are integrated to influence subsequent learning, and the extent to which PreL, IL and other prefrontal areas might be involved remain to be determined.

Reviewer 2:

Asked for anterior commissure figures for PreL-> IL and IL->PreL projection labeling dataset.

The panels for Anterior Commissure (no labeled axons) are now included in Fig. 1.